# A WATERMARK FOR BLACK-BOX LANGUAGE MODELS

## ABSTRACT

Watermarking has recently emerged as an effective strategy for detecting the outputs of large language models (LLMs). Most existing schemes require *white-box* access to the model's next-token probability distribution, which is typically not accessible to downstream users of an LLM API. In this work, we propose a principled watermarking scheme that requires only the ability to sample sequences from the LLM (i.e. *black-box* access), boasts a *distortion-free* property, and can be chained or nested using multiple secret keys. We provide performance guarantees, demonstrate how it can be leveraged when white-box access is available, and show when it can outperform existing white-box schemes via comprehensive experiments.

## 1 INTRODUCTION

It can be critical to understand whether a piece of text is generated by a large language model (LLM). For instance, one often wants to know how trustworthy a piece of text is, and those written by an LLM may be deemed untrustworthy as these models can hallucinate. This problem comes in different flavors – one may want to detect whether it was generated by a *specific* model or by *any* model. Furthermore, the detecting party may or may not have white-box access (e.g. an ability to compute log-probabilities) to the generator they wish to test against. Typically, parties that have white-box access are the owners of the model so we refer to this case as *first-party* detection and the counterpart as *third-party* detection.

The goal of watermarking is to cleverly bias the generator so that first-party detection becomes easier. Most proposed techniques do not modify the underlying LLM's model weights or its training procedure but rather inject the watermark during autoregressive decoding at inference time. They require access to the next-token logits and inject the watermark every step of the sampling loop. This required access prevents third-party users of an LLM from applying their own watermark as proprietary APIs currently do not support this option. Supporting this functionality presents a security risk in addition to significant engineering considerations. Concretely, Carlini et al. (2024) showed that parts of a production language model can be stolen from API access that exposes logits. In this work, we propose a watermarking scheme that gives power back to the people — third-party users can watermark a language model given nothing more than the ability to sample sequences from it. Our scheme is faithful to the underlying language model and it can outperform existing white-box schemes.

## 2 RELATED WORK

We cover related work more extensively in the Appendix; we give a brief overview here. Watermarking in the context of generative language models is a relatively new field, building on prior work in linguistic steganography, where specific words in text are altered to encode information. Early schemes, such as Venugopal et al. (2011), focused on machine translation, but interest surged with the works of Kirchenbauer et al. (2023a;b) and Aaronson (2023), which introduced watermarking for LLMs. These schemes, while effective, can introduce some text distortion, though efforts like Aaronson (2023) and Kuditipudi et al. (2023) seek to make watermarking *distortion-free*. Other works, such as Lee et al. (2023) and Zhao et al. (2023), adapt these methods for specific tasks or to improve resistance to adversarial attacks, while Fernandez et al. (2023) explores new detection tests. Black-box watermarking methods like those by Yang et al. (2023) and Chang et al. (2024) attempt synonym substitution or word insertion but face challenges with text distortion. Paraphrasing

and word substitution attacks pose significant threats to watermarking, leading some to propose semantic-based approaches (Liu et al., 2023b; Hou et al., 2023; Ren et al., 2023; Yoo et al., 2023). However, vulnerabilities persist, as shown by works like Krishna et al. (2024), Zhang et al. (2023) and Gu et al. (2023). Lastly, watermarking has also been studied through the lens of cryptography and classical complexity theory (Christ et al., 2023; Christ & Gunn, 2024).

## 3 ALGORITHM

**High-level sketch.** At a high level, our scheme operates autoregressively; each step, we sample multiple generations from the LLM, score each with our secret key, and output the highest scoring one. We do this repeatedly until our stopping condition (e.g. reaching the stop-token or the max length) is met. To determine whether a piece of text was watermarked, we score it using our key — if it's high, it's likely watermarked. We now describe the algorithm more formally.

**Preliminaries.** We begin with some preliminaries. If $F$ is a cumulative distribution function (CDF), we let $F[s]$ (square brackets) refer to a single draw from a pseudorandom number generator (PRNG) for $F$ seeded by integer seed $s$. Let $F_k$ be the CDF for $\sum_{i=1}^{k} X_i$, where $X_i \overset{iid}{\sim} F$. We sometimes abuse notation and treat a distribution as its CDF (e.g. $N(0,1)(2)$ is the standard normal CDF evaluated at 2) and when the context is clear we let $-F$ be the distribution of $-X$ where $X \sim F$. Now, we detail our proposed algorithm, for which pseudocode is provided in Algorithms 1 and 2 (presented in the Appendix).

Let $F$ be a *continuous* CDF of our choosing, $P$ the input prompt, $K$ a secret integer key known only to the watermark encoder and decoder, LM a conditional language model with vocabulary $\mathcal{V}$ of size $V$, and $h$ a cryptographic hash function (e.g. SHA-256) from $\mathbb{Z}^*$ to $\mathbb{Z}$. Let $n$ be the number of tokens (typically 4 or 5) that serves as input to our pseudo-random function. Our PRF $g : \mathcal{V}^* \to \mathbb{R}$ is given by $g(w) = F[h(K|w)]$, where $|$ denotes concatenation.

**Watermark encoding**. We sample $m$ sequences $\{Q_1, \ldots, Q_m\}$, each consisting of at most $k$ tokens from $\text{LM}(\,\cdot\,|\,P;\,k)$. Let $\{(X_1, c_1), \ldots, (X_j, c_j)\}$ be the *unique* sequences along with their counts from $\{Q_i\}$ — for example, the sequence $X_t$ appears $c_t$ times in $\{Q_i\}$. To score each distinct sequence $X_t$, we first extract its $n$-grams as $\{(X_{t,i-n-1}, \ldots, X_{t,i})\}_{i=1}^{|X_t|}$, where we allow the left endpoint to spill over only to earlier-generated tokens and not the original prompt tokens. $l$-grams are taken instead for boundary indices with only $l-1 < n-1$ eligible tokens strictly left of it. We compute an integer seed for each $n$-gram $w$, as $h(K|w)$. Given a collection of seeds with their associated sequences we deduplicate seeds across the collection. We do this by picking one instance of the seed *at random* and remove all remaining instances from the collection. We ensure every sequence has at least one seed by adding a random seed not already used, if necessary. For each sequence $X_t$, we iterate through its new seeds $S_t$ (order does not matter) and compute the quantity $u_t = F_{|S_t|}\left(\sum_{i=1}^{|S_t|} F[S_{t,i}]\right)$. Finally we compute $i^* = \operatorname{argmax}_{i=1}^{j} u_i^{m/c_i}$ and choose $X_{i^*}$ as our watermarked sequence of length at most $k$. To generate longer texts, we run the the aforementioned process iteratively, where we condition the language model on $P$ and the tokens generated thus far.

One may notice that the LLM is expected to return at most $k$ tokens. This choice is made to simplify the analysis. In practice, the API may only return texts, not tokens, with no option to specify max length. The watermarker can generate $n$-grams from the responses however they would like (with custom tokenization or not). Furthermore, there is no constraint on $k$; $k$ can be set adaptively to the max length in each batch of returned responses. The main consideration though is smaller $k$ begets a stronger watermark, so if the adaptive $k$ is too large, detectability will suffer.

**Watermark detection**. We treat detection as a hypothesis test, where the null $\mathcal{H}_0$ is that the query text is *not* watermarked with our scheme and secret key and the alternative $\mathcal{H}_1$ is that it is. While Bayesian hypothesis testing could be used, this would require choosing priors for both hypotheses, which could be challenging and a poor choice could lead to terrible predictions. Let $X$ be the query text. Akin to the encoding process, we extract $W$, the set of *unique* $n$-grams from $X$, permitting smaller one near the left boundary. For each $n$-gram $w_t$ we compute $R_t = F[h(K|w_t)]$. Under $\mathcal{H}_0$ (assuming that the test $n$-grams are independent), $R_t \overset{iid}{\sim} F$, so $\sum_{t=1}^{|W|} R_t \sim F_{|W|}$ giving a

$p$-value $p = 1 - F_{|W|}\left(\sum_{t=1}^{|W|} R_t\right)$. Our detection score $s$ is $1 - p$ (higher means more likely to be watermarked).

Another way to compute a $p$-value is to compute token-level $p$-values and, assuming they are independent, combine them using Fisher's method. This way, $p = 1 - \chi^2_{2|W|}\left(-2\sum_{t=1}^{|W|} \log\left(1 - F(R_t)\right)\right)$. Furthermore, tests that incorporate the alternative distribution can be used — the best example being the likelihood ratio test: $s = \sum_{t=1}^{|W|}\left(\log f_1(R_t) - \log f_0(R_t)\right)$, where $f_0$ and $f_1$ are the densities of $R_t$ under $\mathcal{H}_0$ and $\mathcal{H}_1$ respectively. For some choices of $F$ and under some assumptions, $f_1$ may be written explicitly. In other cases, one can *estimate* $f_1$ by logging values of $R_t$ for the watermarked sequence as the encoding is run live or via simulation and then building a kernel density estimator. We consider these alternative detection strategies later for ablative purposes.

**Recursive watermarking**. Since our scheme requires only a black box that samples sequences, it can be applied iteratively or recursively. Consider the following. User 1 uses User 2's LLM service who uses User 3's LLM service, so on so forth until User $t$. Our scheme allows User $i$ to watermark its service with its secret key $K_i$. Each user can then run detection using its key oblivious to whether other watermarks were embedded upstream or downstream. Furthermore, the users can cooperate in joint detection by sharing only $p$-values without revealing their secret key. This property is valuable in the service oriented architectures of today's technology stack.

Consider the special case that all users are actually the same entity in possession of $t$ distinct keys $\{K_1, \ldots, K_t\}$. Then the iterative watermarking becomes a recursive one, where $K_i$ is used to watermark the result of watermarking with keys $\{K_{i+1}, \ldots, K_t\}$. The entity can run DETECT to get a $p$-value for each key and these $t$ $p$-values can subsequently be combined using Fisher's method. We present this recursive scheme in Algorithm 2.

**White-box watermarking**. In the case of $k = 1$, our scheme can be efficiently run for users who have white-box access — with the next-token distribution in hand, one can sample a large number of candidate tokens without any inference calls to the model.

**Extensions**. We discuss extensions in the Appendix.

## 4 THEORY

Our goal here is to show that our scheme is faithful to the model's next-token distribution and to give detection performance guarantees. All proofs are in the Appendix.

**Theorem 4.1** (Distortion-free property)**.** *Let $X$ be any finite sequence and $P$ any prompt. Let $X_u \sim LM(\,\cdot\mid P)$ be the non-watermarked output of the conditional autoregressive language model. Let $X_w$ be the output of the watermarking procedure (WATERMARK in Algorithm 1, for both recursive and non-recursive settings) for the same prompt and model and any choice of remaining input arguments with the constraint that $F$ is a continuous distribution. Furthermore, assume that the deduplicated seeds (determined by hashing the secret key and $n$-grams) across sequences, are conditionally independent given the counts of the sampled sequences. Then, $\mathbb{P}(X_u = X) = \mathbb{P}(X_w = X)$.*

Theorem 4.1 tells us that sampling tokens using our proposed scheme is, from a probabilistic perspective, indistinguishable from sampling from the underlying model, with the caveat that the unique seed values are conditionally independent given the counts of sequences. If we dismiss hash collisions as very low probability events, then since the key is fixed, this reduces to the assumption that unique $n$-grams across the sampled sequences are independent. How strong of an assumption this is depends on many factors such as $m$, the underlying distribution, and the counts $(c_1, \ldots, c_j)$ themselves. One can construct cases where the assumption is reasonable and others where it is blatantly violated (e.g. if $n$-grams within a sequence are strongly correlated). One direction to making the assumption more palatable is to draw a fresh keys i.i.d. for *each* hash call. This would obviously destroy detectability. As a trade-off, one can leverage a set of secret keys (i.e. by drawing keys uniformly at random from a key set), which may reduce distortion, but will hurt detection as each key in the set needs to be tested against.

**Theorem 4.2** (Lower bound on detection ROC-AUC)**.** *Consider the specific case of using flat (i.e. non-recursive) watermarking with $k = 1$ and $F = U(0, 1)$. Let $s_0$ be the score under null that the $T$*

*test tokens[1],assumed to be independent, were generated without watermarking and $s_1$ be the score if they were. We have the following lower bound on the detector's ROC-AUC.*

$$\mathbb{P}(s_1 \geq s_0) \geq \frac{1}{1 + 1/(3T\lambda^2\alpha^2)}, \text{ where}$$

$$\lambda = \frac{1}{\log(m)} \left( \frac{m}{m+1} - \frac{1}{2} \right) \text{ and } \alpha = \mathbb{E}_c \left[ -\sum_{i=1}^{V} \mathbf{1}[c_i > 0] \frac{c_i}{m} \log \left( \frac{c_i}{m} \right) \right].$$

*$\alpha$ represents the average Shannon entropy in the sampled next-token distribution.*

Theorem 4.2 connects detection performance to the language model's underlying distribution, number of sampled tokens $m$, and number of test samples $T$. More entropy and more test samples guarantee higher performance. When the model is extremely confident, $\alpha \to 0$ and so does our lower bound. Note that because $\alpha$ measures the entropy of the empirical distribution arising from *sampling* tokens, it depends on both the underlying next-token probability distribution as well as $m$. Concretely, when conditioned on the next-token probabilities $p$, $c \sim \text{Multinomial}(m, p)$. The largest $\alpha$ is achieved when the $c_i$'s are 1, which can occur when the underlying distribution is uniform (maximal uncertainty) and/or $m$ is not large. In this case, $\alpha \to \log(m)$ and our bound goes to $1/\left( 1 + 1/\left( 3T \left( \frac{m}{m+1} - \frac{1}{2} \right)^2 \right) \right)$. This quantity has very sharp diminishing returns with respect to $m$, so there may be little value in increasing $m$ beyond a certain point. When $m \to \infty$, the bound goes to $1/(1 + 4/(3T))$, which increases very quickly with $T$. A mere 50 test tokens guarantees at least 97% ROC-AUC. We study the interplay of the various factors on our lower bound more carefully in the Appendix.

The intuitions here carry over to other choices of $F$ and $k > 1$, though formal bounds can be tricky to obtain because of difficulty quantifying the alternative distribution. The null distribution is easy — $p$-values are $U(0, 1)$ under $\mathcal{H}_0$, and as a result, we have a straightforward equality on the false positive rate.

**Theorem 4.3** (False positive rate). *No matter the choice of watermarking settings, assuming that the unique test $n$-grams are independent, we have the following equality on the false positive rate of* DETECT, *using decision threshold $t$.*

$$FPR = \mathbb{P}_{\mathcal{H}_0}(s > t) = 1 - t.$$

*This also holds for* DETECTRECURSIVE *if we further assume the $p$-values across secret keys are independent.*

Selecting distinct independent secret keys $\{K_1, \ldots, K_t\}$ (and ignoring hash collisions that arise across calls to DETECT within DETECTRECURSIVE), will help attain the necessary independence.

Although the alternative score distribution is generally intractable, with the strong assumption that there are no duplicate $n$-grams across the candidate sequences, then for a special choice of $F$, we can write the alternative in closed form and formulate the optimal detection test.

**Theorem 4.4** (Optimal detection for Gamma). *Assume that candidate sequences are unique with length $k$ and that the $n$-grams are independent and contain no duplicates. Suppose we choose $F = -Gamma(1/k, \beta)$ (flat scheme), for any rate parameter $\beta$. Let $F_0 = F$ with pdf $f_0$, $F_1 = -Gamma(1/k, m\beta)$ with pdf $f_1$, and $R$ the PRF values of the $T$ test tokens (unique $n$-grams), assumed to be independent. Then, $\forall i$, $R_i \sim F_0$ under the null that the text was watermarked using our procedure and $R_i \sim F_1$ otherwise. The uniformly most powerful test is the log-likelihood ratio test (LRT) with score*

$$s(R) = \sum_{i=1}^{T} \log \frac{f_1(R_i)}{f_0(R_i)}.$$

*Furthermore, for any decision threshold $t$ on score $s$, we have that:*

$$FPR \text{ (Type-I error)} = \mathbb{P}_{\mathcal{H}_0}(s > t) = Gamma(T/k, \beta)(Q(t)), \text{ and}$$

$$FNR \text{ (Type-II error)} = \mathbb{P}_{\mathcal{H}_1}(s \leq t) = 1 - Gamma(T/k, m\beta)(Q(t)), \text{ where}$$

$$Q(t) = \frac{T \log(m)/k - t}{(m-1)\beta}.$$

---

[1]more precisely, $T$ *unique $n$-grams*

In the Appendix, we use Theorem 4.4 to study the impact of $k$, $m$, and $T$ on TPR at fixed FPR. For example, with $T = 100$, $k = 50$, $m = 64$, $\beta = 1$, we can achieve $99.9\%$ TPR at $1\%$ FPR.

For other choices of $F$, we can estimate $f_1$ via simulation. If we assume candidate sequences have the same length $k$ with no duplicate $n$-grams, then we can fill an $m \times k$ matrix with i.i.d. draws from $F$ and pick the first element of the row with the largest row-sum (among the $m$). We do this until we have sufficiently large (e.g. 10,000) samples from $f_1$. We apply a Gaussian kernel-density estimator where the bandwidth is chosen using Scott's rule (Scott, 2015) to estimate $f_1(r)$ for test value $r$. Despite having $f_0$ in closed-form, for consistency, we can also estimate it non-parametrically by drawing from $F$.

## 5 EXPERIMENTS

In this section, we compare the performance of our scheme with that of prior work.

### 5.1 MODELS, DATASETS, AND HYPER-PARAMETERS

**Models and Datasets**. Our main model and dataset is the MISTRAL-7B-INSTRUCT (Jiang et al., 2023) hosted on Huggingface[2] with bfloat16 quantization, and *databricks-dolly-15k*[3] (Conover et al., 2023), an open source dataset of instruction-following examples for brainstorming, classification, closed QA, generation, information extraction, open QA, and summarization. We use prompts from the brainstorming, generation, open QA (i.e. general QA), and summarization categories, whose human responses are at least 50 tokens long (save one example, which was removed because the prompt was extremely long). For each of the 5233 total prompts, we generate two non-watermarked responses — a stochastic one using temperature 1, and the greedy / argmax decoding — along with a watermarked one for each scheme. We always force a minimum (maximum) of 250 (300) new tokens by disabling the stop token for the first 250 tokens, re-enabling it, and stopping the generation at 300, regardless of whether the stop token was encountered. To simulate real-world use, we de-tokenize the outputs to obtain plain text, and re-tokenize them during scoring. We study performance as a function of token length $T \leq 250$ by truncating to the first $T$ tokens.

For completeness, we also present the key results when GEMMA-7B-INSTRUCT[4] with bfloat16 quantization is applied to the test split of *eli5-category*[5]. Prompts are formed by concatenating the the *title* and *selftitle* fields. Only examples with non-empty *title* and whose prompt contains a *?* are kept — for a total of 4885 examples.

**Hyper-parameters**. We consider the following choices of CDFs $F$ / $F_k$. **(1)** $F = U(0, 1)$ and $F_k = \text{IrwinHall}(k)$. **(2)** $F = N(0, 1)$ and $F_k = N(0, k)$. **(3)** $F = -\text{Gamma}(1/k, 1)$ and $F_k = -\text{Exp}(1)$. **(4)** $F = \chi_2^2$ and $F_k = \chi_{2k}^2$.

### 5.2 EVALUATION METRICS

We evaluate performance using three criteria.

**Detectability**. How well can we discriminate between non-watermarked and watermarked text? We choose non-watermarked text to be text generated by the same model, just without watermarking applied during decoding. There are three reasons for choosing the negative class in this way. Firstly, it makes controlling for text length easier as we can generate as many tokens as we do for watermarked samples — in contrast, human responses are of varying lengths. Secondly, watermarked text has far more token / $n$-gram overlap with its non-watermarked counterpart than the human reference, which makes detection more challenging. Lastly, since one intended use case of our scheme is for third-party users of a shared LLM service, users may want to distinguish between their watermarked text and non-watermarked text generated by the same LLM service.

Our primary one-number metric is ROC-AUC for this balanced binary classification task. Since performance at low FPR is often more useful in practice, we report the partial ROC-AUC (pAUC)

[2]https://huggingface.co/mistralai/Mistral-7B-Instruct-v0.1

[3]https://huggingface.co/datasets/databricks/databricks-dolly-15k

[4]https://huggingface.co/google/gemma-7b-it

[5]https://huggingface.co/datasets/rexarski/eli5_category

for FPR $\leq$ a target FPR (taken to be 1%), which we find to be more meaningful than TPR at the target FPR. We look at performance as a function of length by truncating the positive and negatives samples to lengths $\{25, 50, 75, 100, 150, 200, 250\}$. To understand aggregate performance, we pool all different length samples together and compute one ROC-AUC. Here, it is paramount that the detection score be length-aware to ensure that a single decision threshold can be used across lengths.

**Distortion**. Our scheme, along with most of the baselines, boasts a *distortion-free* property. This property comes with assumptions that are often violated in practice, for example by reuse of the secret key across watermarking calls. We quantify how *faithful* the watermarking procedure is to the underlying generative model by computing both the *perplexity* and *likelihood* of watermarked text under the generator (without watermarking). We include likelihood as the log-probabilities used in calculating perplexity can over-emphasize outliers.

**Quality**. Watermarking may distort the text per the model, but does the distortion tangibly affect the *quality* of the text? Quality can be challenging to define and measure — one proxy is likelihood under a much larger model than the generator. Alternatively, one can run standard benchmark NLP tasks and use classic metrics like exact match, etc. We instead opt for using Gemini-1.5-Pro as an LLM judge and compute pairwise win rates for each watermark strategy against no watermarking (greedy decoding). We do this in two ways for each scheme — (1) we compute win rates using a single response for each prompt and (2) we first ask the LLM judge to pick the best of 3 responses for each prompt and compute win rates using the best response. (2) represents the common practice of sampling a few generations from the LLM and selecting the best one using some criterion. It captures diversity, as methods that can express an answer in a few different good ways will have an advantage. A caveat with win rates is that they may not reflect the *degree* by which one method is better or worse. For instance, if one strategy's output was always *marginally* worse than no watermarking, the win rate would be 0% — the same as if it were *much* worse.

### 5.3 ADVERSARIAL ATTACKS

An adversary in possession of watermarked text (but who lacks knowledge of the secret key) may try to evade detection. We study how detectability degrades under two attack strategies —- *random token replacement* and *paraphrasing*.

**Random token replacement**. Here, we take the watermarked tokens and a random $p$-percent them are corrupted by replacing their token with a random different one. $p$ is taken to be $[10, 20, 30, 40, 50]$. This attack strategy is cheap for the adversary to carry out but will significantly degrade the quality of the text.

**Paraphrasing**. In this attack, the adversary attempts to evade detection by paraphrasing the watermarked text using the model. We use Gemini-1.5-Pro to paraphrase each non-truncated watermarked generation. Details are deferred to the Appendix.

### 5.4 BASELINES

The watermark schemes we consider here operate token-by-token in the autoregressive decoding loop. Let $p$ be the next-token probability distribution. Higher detection scores indicate higher confidence that the query text is watermarked.

**Aaronson (A)**. Aaronson (2023) computes a PRN for each token $i$ in the vocabulary as $u_i = U(0, 1)[h(i|w|K)]$, where $w$ is preceding $(n-1)$-gram, $K$ is the secret key and $h$ is a cryptographic hash. Token $i^*$ is selected, where $i^* = \operatorname{argmax}_i u_i^{1/p_i}$. At test time, $n$-grams $\{w_i\}_{i=1}^T$ are extracted from the query test and the detection score $s$ is $-\sum_{i=1}^T \log(1 - R_i)$, where $R_i = U(0, 1)[h(w_i|K)]$. $n$ is set to 4. This choice strikes a good balance between generation quality / diversity and robustness to attacks. The scheme boasts a *distortion-free* property, but the generated text is a deterministic function of the prompt — i.e. only one generation is possible conditioned on a particular prompt.

**Remark.** *If $k = 1$ and $F = U(0, 1)$, then our watermark encoding can be viewed as a stochastic version of Aaronson (2023)'s. As $m \to \infty$, $c_t/m \overset{a.s.}{\to} p_t$, where $p_t$ and $c_t$ are the probability and observed occurrences of token $t$.*

| | PPL | WR | WR (3) | AUC | pAUC | C. AUC | C. pAUC | P. AUC | P. pAUC |
|---|---|---|---|---|---|---|---|---|---|
| Max Std. Error | 0.03 | - | - | 0.1 | 0.3 | 0.2 | 0.3 | - | - |
| Greedy Decoding | 1.37 | - | - | - | - | - | - | - | - |
| Random Sampling | 3.50 | 49.6 | 65.3 | - | - | - | - | | |
| Aaronson | 2.81 | 45.3 | 45.3 | 71.7 | 65.5 | 65.6 | 60.3 | 53.9 | 50.5 |
| Aaronson Cor. | 2.81 | - | - | 97.9 | 83.6 | 94.8 | 73.2 | 58.8 | 50.7 |
| Kuditipudi | 3.55 | 50.3 | 67.3 | 87.8 | 76.6 | 87.2 | 74.4 | 75.9 | 53.2 |
| Kirchenbauer 0.5 | 3.39 | 49.6 | 66.6 | 73.2 | 52.0 | 71.0 | 51.4 | 49.0 | 49.8 |
| 1 | 3.37 | 50.1 | 67.0 | 86.9 | 60.6 | 83.7 | 57.1 | 52.9 | 49.9 |
| 2 | 3.69 | 47.9 | 64.1 | 97.0 | 83.3 | 95.4 | 77.4 | 58.4 | 50.3 |
| 3 | 4.67 | 41.5 | 58.4 | 99.3 | 94.4 | 98.6 | 90.9 | 63.4 | 51.5 |
| 4 | 5.81 | 26.0 | 41.2 | 99.8 | 98.4 | 99.6 | 96.8 | 66.4 | 52.7 |
| Flat ($k=1$) 2 | 3.46 | 50.0 | 66.4 | 90.2 | 68.8 | 82.0 | 58.7 | 50.5 | 50.3 |
| 4 | 3.36 | 50.8 | 67.0 | 95.8 | 82.9 | 90.3 | 70.5 | 51.3 | 50.6 |
| 16 | 3.20 | 47.7 | 64.5 | 97.7 | 89.7 | 93.9 | 79.1 | 52.7 | 51.1 |
| 32 | 3.06 | 48.4 | 65.3 | 97.8 | 90.2 | 94.2 | 80.0 | 53.0 | 50.8 |
| 512 | 2.63 | 47.7 | 62.5 | 97.7 | 90.0 | 94.1 | 79.7 | 54.6 | 51.3 |
| 1024 | 2.61 | 47.7 | 62.2 | 97.7 | 90.0 | 94.0 | 79.7 | 52.8 | 51.1 |
| Flat ($k=10$) 2 | 4.10 | 46.1 | 62.2 | 83.4 | 55.8 | 73.6 | 52.0 | 49.0 | 50.0 |
| 4 | 4.06 | 45.2 | 61.5 | 93.8 | 72.7 | 85.7 | 59.4 | 51.3 | 50.3 |
| 16 | 3.86 | 44.6 | 60.6 | 97.8 | 87.0 | 93.1 | 73.5 | 54.3 | 50.7 |
| 32 | 3.80 | 43.0 | 60.8 | 98.2 | 89.0 | 94.0 | 76.7 | 55.0 | 50.8 |
| Flat ($k=50$) 2 | 3.79 | 48.5 | 64.2 | 69.6 | 50.7 | 62.2 | 50.3 | 47.0 | 50.0 |
| 4 | 3.76 | 47.7 | 63.9 | 82.9 | 53.5 | 71.9 | 51.3 | 49.4 | 50.0 |
| 16 | 3.72 | 48.3 | 64.2 | 92.7 | 66.7 | 83.1 | 55.6 | 50.5 | 50.1 |
| 32 | 3.67 | 47.3 | 63.9 | 94.2 | 71.6 | 85.5 | 58.1 | 51.1 | 50.5 |
| Rec. ($k=1$) 4 | 3.41 | 49.0 | 65.0 | 93.4 | 75.5 | 86.3 | 63.2 | 48.4 | 50.4 |
| 16 | 3.33 | 49.2 | 66.2 | 95.4 | 82.9 | 90.6 | 71.8 | 53.4 | 50.8 |
| 32 | 3.29 | 48.4 | 64.3 | 96.3 | 85.0 | 91.6 | 73.5 | 49.4 | 50.8 |
| 512 | 3.05 | 48.3 | 64.5 | 97.2 | 87.9 | 92.6 | 76.5 | 50.4 | 51.2 |
| Rec. ($k=10$) 4 | 4.13 | 45.7 | 61.3 | 88.6 | 61.7 | 78.8 | 53.8 | 48.0 | 50.0 |
| 16 | 4.13 | 43.7 | 59.7 | 93.4 | 74.0 | 86.8 | 61.6 | 52.9 | 50.4 |
| 32 | 4.06 | 42.9 | 59.5 | 94.8 | 76.9 | 88.1 | 63.2 | 50.6 | 50.3 |
| Rec. ($k=50$) 4 | 3.79 | 48.2 | 63.8 | 74.2 | 51.2 | 65.1 | 50.5 | 46.5 | 49.9 |
| 16 | 3.77 | 47.0 | 64.0 | 81.2 | 54.5 | 73.3 | 51.9 | 51.4 | 50.2 |
| 32 | 3.79 | 47.2 | 63.3 | 83.3 | 55.7 | 74.4 | 52.2 | 49.4 | 50.0 |

Table 1: Main table of results, showing our black-box scheme and its recursive variant for various $k$'s and $m$'s, along with baselines. PPL, WR and WR (3) refer to perplexity, win rate of a single response, and win rate of the best-of-3 responses respectively. pAUC is ROC-AUC up to max FPR of 1%. C and P stand for 10% corruption and paraphrasing attack. For paraphrasing, target lengths of [150, 200, 250] are used in the AUC / pAUC computation here and elsewhere as performance is essentially random on shorter lengths. The standard errors are quite small and the maximum across rows is shown for each column. AUCs and pAUCS and their standard errors are scaled by 100.

**Aaronson Corrected (AC).** Aaronson (2023)'s detection score $s_A$ is not length-aware and consequently a single decision threshold across scores involving various lengths results in poor performance, as we later show. Observing that $s_A$ is a sum of log $p$-values, $s_A \sim \text{Gamma}(T, 1)$, or equivalently, $2s_A \sim \chi^2_{2T}$ under the null that all test tokens are non-watermarked. We propose the new *corrected* detection score, $s = 1 - \text{Gamma}(T, 1)(s_A) = 1 - \chi^2_{2T}(2s_A)$. For completeness we also experiment with a $p$-value computed in the way we do for our method — concretely as, $1 - \text{IrwinHall}(T) \left( \sum_{i=1}^{T} R_i \right)$. Note that both transformations are monotonic so they have no effect on ROC-AUC when T is *fixed*.

**Kirchenbauer (KB).** Kirchenbauer et al. (2023a) uses the current $n$ previous tokens to pseudorandomly partition the vocabulary for the next token into two lists: a green list of size $\gamma V$ and a red list

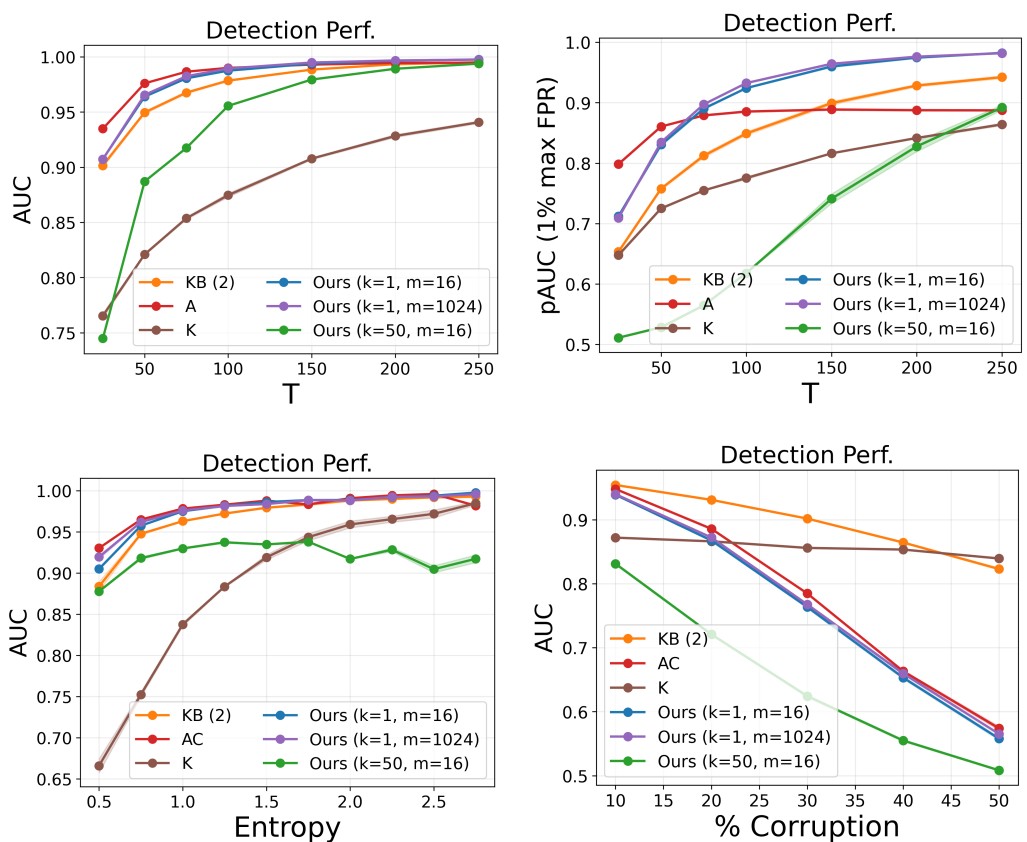

Figure 1: **Top**: Detection AUC and pAUC with 1% max FPR for a range of target text lengths when there is no corruption. **Bottom Left**: AUC (mixed $T$'s) as a function of the average non-watermarked response entropy of the examples used in the calculation. $x$-coordinate $x$ corresponds to the bucket of examples whose entropy is between $[x - 0.25, x]$ nats. **Bottom Right**: Effect of amount of random token corruption on AUC (mixed $T$'s).

consisting of the remainder. A positive bias of $\delta$ is added to the logits of the green list tokens while those of the red list are left unchanged. This has the effect of modifying $p$ so that green list tokens are more probable. The score for a text consisting of $T$ tokens, $T_g$ of which were found to be green is, $s = (T_g - \gamma T)/\sqrt{T\gamma(1 - \gamma)}$. We incorporate the latest updates to the algorithm[6], such as including the current token in the $n$-gram and skipping duplicate $n$-grams at test time. We set $n = 4$, $\gamma = 0.25$, and $\delta \in \{0.5, 1, 2, 3, 4\}$.

**Kuditipudi (K)**. A drawback of using the last $n$ tokens as a basis for the PRF is that changing just one of them changes the output and hurts detection. Kuditipudi et al. (2023) addresses this limitation as follows. Consider a secret, finite ordered list of seeds of length $k$. Start watermarking by selecting a position in the seed list uniformly at random and apply the selection rule of Aaronson (2023) with the PRNG seeded to the current value. Advance to the next seed in the list (wrap-around if you are at the end) and repeat. Scoring is done by conducting a permutation test evaluating how compatible the query text is with the specific list of seeds used during encoding as opposed to any other random list of seeds of the same length. As the random starting position is not known during scoring, an alignment score based on the Levenshtein distance is given that considers alignments of various subsequences of the text and seeds. The proposed method is quite similar to Aaronson (2023) with the difference of using a fixed list of seeds (instead of context tokens to determine the seed) and using a permutation test for scoring. The upside is robustness to token substitution attacks; the downside is significantly higher computational cost for scoring. Larger $k$ offers more diversity and quality in generation but comes with costlier and weaker detection. The scheme is distortion-free. Following

---

[6] https://github.com/jwkirchenbauer/lm-watermarking

their work, we let $k = 256$ and accelerate the permutation test by pre-computing 5000 reference values for the secret list using snippets from the train set of *C4-realnewslike* (Raffel et al., 2019) at the various target lengths we evaluate on.

## 5.5 EXPERIMENTAL RESULTS

Table 1 shows results for baselines and our scheme using $F = U(0, 1)$ and $p$-values for scoring, as detailed in Algorithms 1 and 2. For the recursive scheme, depth is $\lg(m)$ (i.e. $m = 2$ for each imaginary watermarker). Here, the negative class is non-watermarked argmax/greedy generations. Results for using stochastic (temperature 1) generations as the negative as well as the average likelihood scores are presented in Table 5 (Appendix); the trends remain the same. We summarize our observations on MISTRAL-7B-INSTRUCT on *databricks-dolly-15k*, which also hold for GEMMA-7B-INSTRUCT on *eli5-category* (presented in the Appendix).

### 5.5.1 OVERALL PERFORMANCE OF OUR FLAT AND RECURSIVE SCHEMES

**Our scheme is a competitive option for white-box watermarking**. Is it better to use our method or alternatives in the white-box setting? When $k = 1, m = 1024$, we are able to achieve better perplexity (2.61 vs. 2.81), better diversity (62.2% vs. 45.3% on best-of-3 win rates) and comparable detection performance than Aaronson (2023). Furthermore, it has better perplexity (2.61 vs. 3.55) and detection performance (97.7% vs. 87.8% AUC) than Kuditipudi et al. (2023). By cranking up $\delta$, Kirchenbauer et al. (2023a) can achieve strong detection but at the expense of perplexity. When matched on perplexity, we achieve better detection. For example, $\delta = 0.5$ achieves 3.39 PPL and 73.2% AUC compared to our 2.61 PPL and 97.7% AUC. GEMMA-7B-INSTRUCT on *eli5-category* with $k = 1, m = 1024$ outperforms Kuditipudi et al. (2023) and is on-par with Aaronson (2023) (see Appendix). Kirchenbauer et al. (2023a) with $\delta = 0.5$ gives 1.649 PPL and 61.6% AUC whereas $k = 1, m = 1024$ gets us 1.610 PPL with 93.2% AUC and *even* 1.645 PPL with 89.7% AUC when $k = 50, m = 16$ (black-box).

**Flat watermarking outperforms recursive.** Across metrics and settings we see that the flat scheme outperforms its recursive counterpart, suggesting it is more effective when a strong signal is embedded using a single key rather than when multiple weak signals are embedded with different keys. For example, when $k = 1, m = 32$ flat (recursive) PPL and AUC are 3.06 (3.29) and 97.8% (96.3%) respectively.

### 5.5.2 EFFECTS OF HYPERPARAMETERS

**Increasing $m$ improves perplexity but hurts diversity.** Across $k$'s, we observe that perplexity decreases as $m$ increases, but that win rates, especially when best-of-3 generations are used, decrease. For example, when $k = 1$, increasing $m$ from 2 to 1024 decreases perplexity from 3.46 to 2.61 but also drops the best-of-3 win rate from 66.4% to 62.2%. As remarked earlier, as $m \to \infty$, $c_t/m \to p_t$ and our scheme becomes less diverse — deterministic conditioned on the prompt, like Aaronson's. On the flip side, large $m$ reduces sampling noise which drives down perplexity.

**Increasing $m$ improves detection but has diminishing returns.** Across the board we see that detection improves as $m$ increases, but there are diminishing returns. For example, when $k = 1$, our AUC increases from 90.2% to 95.8% as $m$ goes from 2 to 4, but flattens out when $m$ hits 16. This corroborates our theoretical intuition from Theorem 4.2 which is further explored in Figure 4 (Appendix).

**For fixed $m$, increasing $k$ hurts detection performance.** For fixed $m$ and target generation length $T$, increasing $k$ gives us fewer opportunities (fewer calls to WATERMARKSINGLE) to inject the watermark signal, and detection consequently suffers. For example, when $m = 32$, AUC drops from 97.8% to 94.2% when $k$ increases from 1 to 50.

$U(0, 1)$ **slightly outperforms alternative distributions. Flat distributions may offer better robustness to attacks.** In Table 4 (Appendix), we see that $U(0, 1)$ fares comparably to $N(0, 1)$ and slightly outperforms $\chi_2^2$ both on detection and perplexity. For example, when $k = 50, m = 2$, $U(0, 1)$ and $\chi_2^2$ have AUCs of 69.6% and 68.1% respectively. Furthermore, we find evidence that $U(0, 1)$ offers better protection to attacks. For example, when $k = 50, m = 32$, the AUC for $U(0, 1)$ ($\chi_2^2$) degrades from 94.2% (94.5%) to 85.5% (84.5%) in the presence of 10% random token

corruption. We provide some intuition for why flat distributions like $U(0, 1)$ may be more robust than those with quickly decaying tails. Consider shaping the continuous $F$ so it approaches $\text{Bern}(p)$ (i.e. $f(x) \approx (1-p)\delta(x) + p\delta(x-1)$), where $p$ is very small. Suppose $k$ is large and $m$ is small. Then, the winning sequence $X_{i*}$ will have extremely few (if any) of its $R_j$'s equal to 1. If the text is unmodified and these few $n$-grams are kept intact, we are fine, but if they are corrupted in an attack, then the watermarking signal is effectively lost. In other words, flat distributions *smear* the watermarking signal over more tokens than do sharper distributions, which localize the signal to few lucky token positions. However, whereas scoring with $F_T = \text{IrwinHall}(T)$ when $F = U(0, 1)$ involves computing $T$-fold convolutions or cardinal B-splines, when $F = N(0, 1)$, $F_T$ is easier to compute for very large $T$; specifically, $F_T(x) = F\left(x/\sqrt{T}\right)$.

### 5.5.3 OBSERVATIONS ON DETECTION

**Length correction of Aaronson (2023) is crucial.** Recall that the ROC-AUCs presented in Table 1 are computed over a pool of different lengths. Our $p$-value-based score for Aaronson (2023) improves detection significantly; for example, AUC goes from 71.7% to 97.9%. Table 7 (Appendix) shows that the *sum-based* $p$-value correction fares a bit worse, which was a little surprisingly given that this worked the best for our scheme, even for the $k = 1$ case.

**Sum-based $p$-values outperform Fisher ones.** In Table 2 (Appendix), we observe that replacing our *sum-based* $p$-value (where $\mathcal{H}_0$ is that $\sum_i^T r_i \sim F_T$) by a Fisher combination of token-level $p$-values, hurts detection performance. For example, when $k = 1, m = 2$, AUC degrades from 90.2% to 86.3%. Note that when $k = 1$, this setting corresponds exactly to a stochastic version of Aaronson (2023).

**Likelihood-ratio scoring does well for large $k$ and small $m$, when its assumptions are more realistic.** In Table 3 (Appendix), we observe that likelihood-based scoring — both when the distribution is Gamma and the exact likelihood ratio test (LRT) is used and under KDE with alternative distributions — performs the best when the assumptions of no duplicate sequences or $n$-grams hold better. This happens when the sequences are long (large $k$) and when fewer sequences are sampled (small $m$). For example, when $k = 1$, AUC *degrades* monotonically from 78.1% to 55.4% as $m$ increases from 2 to 1024. In contrast, AUC under the $p$-value-based scoring *increases* monotonically with $m$, from 90.2% to 97.7%. Larger $m$ increases the number of duplicate sequences sampled, increasing the importance of the latent exponent $m/c_i$ used in the scoring and deviating us further from the LRT assumptions. However, LRT has the potential to be an effective alternative when $k$ is large. For example, when $k = 50$ and $m = 32$, Uniform KDE-based LRT gives AUC of 95.5% compared to $p$-value's 94.2%.

**Detection performance improves sharply with test samples $T$.** Figure 1 shows the effect of $T$ on AUC. We see sharp improvements w.r.t. to $T$, even when $k$ is large and $m$ is small, highlighting the power of more test samples to counteract a weaker watermark signal.

**Entropy improves detection performance.** In Figure 1 we bucket prompts based on the entropy of their non-watermarked response and then look at detection AUC on samples in each bucket. As we expect, detection improves when the prompts confer more entropy in the response. This trend is more stark for our method.

**Paraphrasing can be extremely effective at destroying watermarks.** We observe that paraphrasing can effectively erase the watermark as detection performance for most methods is near random. Kuditipudi et al. (2023) and Kirchenbauer et al. (2023a) with large $\delta$ do better on AUC (but not so much on pAUC). Furthermore, in Figures 1 and 2 (Appendix) we observe that large amounts of random token corruption hurts our scheme and Aaronson (2023)'s more than it that of Kirchenbauer et al. (2023a) or Kuditipudi et al. (2023).

## 6 CONCLUSION

In this work, we present a framework for watermarking language models that requires nothing more than a way to sample from them. Our framework is general and extensible, supporting various real world use-cases, including the setting where the next-token probabilities are in fact available. We study its various components and the trade-offs that arise, provide formal guarantees for the theoretically-inclined as well as concrete recommendations for the practitioner.

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

## A  APPENDIX

### A.1  ALGORITHM

---

**Algorithm 1** Black-Box Watermarking

---

1: **function** WATERMARK(cdf $F$, key $K$, # cand $m$, ctx len $n$, prompt $P$, seq len $k$, LM)
2:    $O \leftarrow \phi$
3:    **while** $\neg$ STOPCOND($O$) **do**       $\triangleright$ *Continue until stop token is encountered or max length reached.*
4:      $O \leftarrow O \mid$ WATERMARKSINGLE($F, K, m, n, P|O, k$, LM)
5:    **return** $O$

6: **function** WATERMARKSINGLE(cdf $F$, key $K$, # cands $m$, ctx len $n$, prompt $P$, seq len $k$, LM)
7:    $Q_1, \ldots, Q_m \sim$ LM$(\cdot \mid P; \ k)$       $\triangleright$ *Draw $m$ sequences from LM, each with at most $k$ tokens.*
8:    $(X_1, c_1), \ldots, (X_j, c_j) \leftarrow$ UNIQUESEQSWITHCOUNTS($(Q_1, \ldots, Q_m)$)
9:    $u_1, \ldots, u_j \leftarrow$ SCORESEQS($F, (X_1, \ldots, X_j), K, n, P$)
10:    $i^* \leftarrow \text{argmax}_{i=1}^{j} u_i^{m/c_i}$
11:    **return** $X_{i^*}$

12: **function** SCORESEQS(cdf $F$, candidates $C$, key $K$, ctx len $n$, prefix $P$)
13:    $Z \leftarrow \phi$
14:    **for** $X_i$ in $C$ **do**
15:      **for** $w$ in NGRAMS $(X_i, n, P)$ **do**       $\triangleright$ *Don't compute $n$-grams over original prompt.*
16:        $Z \leftarrow Z \mid (i, \text{INTHASH}(K|w))$       $\triangleright$ *Apply cryptographically secure integer hash.*
17:    $Z \leftarrow$ REMOVEDUPLICATES($Z$)
18:    **for** $i, S$ in SORTEDGROUPBY($Z$) **do**       $\triangleright$ *Iterate through each candidate's set of unique seeds.*
19:      $R \leftarrow (F[s]$ for $s$ in $S)$
20:      $u_i \leftarrow F_{|R|}\left(\sum_j R_j\right)$
21:    **return** $u_1, \ldots, u_{|C|}$

22: **function** DETECT(cdf $F$, tokens $X$, key $K$)       $\triangleright$ *p-value-based detection.*
23:    $S \leftarrow \phi$
24:    **for** $w$ in NGRAMS $(X, n, \phi)$ **do**
25:      $S \leftarrow S \mid$ INTHASH($K|w$)
26:    $S \leftarrow$ REMOVEDUPLICATES($S$)
27:    $R \leftarrow (F[s]$ for $s$ in $S)$
28:    **return** $F_{|R|}\left(\sum_j R_j\right)$       $\triangleright$ *Higher score means higher likelihood of being watermarked.*

---

**Algorithm 2** Recursive Black-Box Watermarking

---

1: **function** WATERMARKRECURSIVE($F, (K_1, \ldots, K_t), m^t, n, P, k$, LM) $\triangleright$ *Sub. for* WATERMARKSINGLE.
2:    **if** $t = 1$ **then**
3:      $M =$ LM$(\cdot \mid \cdot; \cdot)$
4:    **else**
5:      $M =$ WATERMARKRECURSIVE($F, (K_2, \ldots, K_t), m^{t-1}, n, \cdot, \cdot,$ LM)
6:    **return** WATERMARKSINGLE($F, K_1, m, n, P, k, M$)

7: **function** DETECTRECURSIVE(cdf $F$, tokens $X$, keys $(K_1, \ldots, K_t)$)
8:    $P \leftarrow \phi$
9:    **for** $K_i$ in $(K_1, \ldots, K_t)$ **do**
10:      $P \leftarrow P \mid (1 - $ DETECT($F, X, K_i$))
11:    $y \leftarrow -2 \sum_i \log P_i$       $\triangleright$ *Combine p-values using Fisher's method.*
12:    **return** $\chi_{2t}^2(y)$

---

### A.2  EXTENSIONS

We now discuss extensions of our method. At its crux, the scheme samples sequences of text from a service, divides each unique sequence into a bag of units (namely $n$-grams) where each unit is scored

using a PRF and the scores are combined in an order-agnostic way. The strength of the watermark depends on the number of *distinct* units across the candidate sequences and the robustness depends on how many of the units are kept intact after the attack. Although any symmetric monotone function can be used instead of the simple summation of the PRNs for each unit, we do not see any compelling reason to make our algorithm more general in this way. However, we briefly highlight some other possible extensions.

*Beam search.* Rather than drawing i.i.d. samples from the model, one can apply our watermark selection to the sequences that arise from beam search, with the caveat that this would violate our distortion-free property.

*Semantic watermarking.* Rather than use $n$-grams, the watermarker can extract a set of meaningful semantic units for each sampled text. Robustness may be improved as these units will largely remain intact under an attack like paraphrasing. On the other hand, many of the sampled sequences will have the same *meaning*, so there may be a lot of duplicate units across the candidate sequences, which would degrade the watermark strength.

*Paraphrasing.* Thus far, we assumed the service provides $m$ draws from the LLM. If $m$ is large, this can be prohibitively expensive. The resource-constrained may consider the following alternative: draw one sample from the LLM and feed it to a much cheaper paraphrasing model to generate $m$ paraphrases. The downside is that there may be a lot of duplicate $n$-grams across the candidate set.

## A.3 FULL RELATED WORK

Watermarking outside of the context of generative LLMs, which is sometimes referred to as linguistic steganography, has a long history and typically involves editing specific words from an non-watermarked text. Watermarking in the modern era of generative models is nascent — Venugopal et al. (2011) devised a scheme for machine translation, but interest in the topic grew substantially after the more recent seminal works of Kirchenbauer et al. (2023a;b) and Aaronson (2023). Many effective strategies employ some form of pseudorandom functions (PRFs) and cryptographic hashes on token $n$-grams in the input text. Kirchenbauer et al. (2023a) proposes modifying the next-token probabilities every step of decoding such that a particular subset of the vocabulary, referred to as *green list tokens*, known only to those privy to the secret key, are made more probable. Watermarked text then is expected to have more green tokens than non-watermarked text and can be reliably detected with a statistical test. The scheme distorts the text, but with the right hyper-parameters a strong watermark may be embedded with minimal degradation in text quality.

Meanwhile, Aaronson (2023) proposes a clever *distortion-free* strategy which selects the token that is both highly probable and that achieves a high PRF value. Kuditipudi et al. (2023) applies a scheme similar in spirit to Aaronson (2023) but to improve robustness to attacks, pseudorandom numbers (PRNs) are determined by cycling through a fixed, pre-determined sequence of values called the *key*, rather than by $n$-grams. They compute a $p$-value using a permutation test to determine if the text was watermarked with *that specific* key.

Lee et al. (2023) adapts Kirchenbauer et al. (2023a)'s scheme for code-generation by applying the watermark only at decoding steps that have sufficient entropy. Zhao et al. (2023) investigates a special case of Kirchenbauer et al. (2023a) where $n = 0$ for improved robustness to adversarial corruption. Fernandez et al. (2023) tests various watermarking schemes on classical NLP benchmarks and also introduces new statistical tests for detection — most notably, they suggest skipping duplicate $n$-grams during testing.

Yang et al. (2023) introduces a scheme that relies on black-box access to the LLM. Their method samples from the LLM and injects the watermark by replacing specific words with synonyms. Although their approach shares the assumption of black-box LLM access, as in our work, it has limitations not present in ours: the watermarking process is restricted to words that can easily be substituted with multiple synonyms, synonym generation is powered by a BERT model (Devlin, 2018), making it computationally expensive, and the scheme is not distortion-free. Chang et al. (2024) presents POSTMARK, a black-box watermarking method that uses semantic embeddings to identify an input-dependent set of words. These words are then inserted into the text by an LLM after decoding. However, this approach is also not distortion-free, as the insertion of words by the LLM often results in significantly longer watermarked text.

Given the weakness of many schemes to paraphrasing or word substitution attacks, some have proposed watermarking based on semantics and other features that would remain intact for common attack strategies (Liu et al., 2023b; Hou et al., 2023; Ren et al., 2023; Yoo et al., 2023). Meanwhile, others have viewed the problem through the lens of cryptography and classical complexity theory (Christ et al., 2023; Christ & Gunn, 2024). Lastly, Liu et al. (2023a) proposes an un-forgeable publicly verifiable watermark algorithm that uses two different neural networks for watermark generation and detection. Huang et al. (2023) improves the statistical tests used for detection, providing faster rates than prior work.

As the deployment of watermarks to LLMs is still early and also presumably secretive, the correct threat model is still undetermined. Krishna et al. (2024) shows that paraphrasing can evade both third-party and watermarking detectors alike. Some may posit that attacks like paraphrasing or round-trip translation are unrealistic since either they are too expensive to conduct at scale or parties in possession of a capable paraphrasing model have adequate resources to serve their own LLM. Zhang et al. (2023) show that attackers with weaker computational capabilities can successfully evade watermarks given access to a *quality oracle* that can evaluate whether a candidate output is a high-quality response to a prompt, and a *perturbation oracle* which can modify an output with a non-trivial probability of maintaining quality. Alarmingly, Gu et al. (2023) demonstrates that watermarks can be learned — an adversary can use a teacher model that employs decoder-based watermarking to train a student model to emulate the watermark. Thibaud et al. (2024) formulates tests to determine whether a black-box language model is employing watermarking, and they do not find strong evidence of watermarking among currently popular LLMs.

### A.4 ADDITIONAL EXPERIMENTAL DETAILS

**Prompting strategies for Gemini.** We use Gemini for paraphrasing and as an LLM judge. Occasionally, Gemini will refuse to return a response due to safety filters that cannot be bypassed. We use the following prompt to compute win rates:

*"Is (A) or (B) a better response to PROMPT? Answer with either (A) or (B). (A): GREEDY RESPONSE. (B): WATERMARKED RESPONSE."*

For determining the best response, we use:

*"Is (A), (B), or (C) the best responses to PROMPT? Answer with either (A), (B), (C). (A): RESPONSE 1. (B): RESPONSE 2. (C): RESPONSE 3."*

In both cases, we search for the first identifier (i.e. "(A)", "(B)", "(C)"). If one is not found or if Gemini does not return a response, the example is not used in the win rate calculation or the first response is chosen.

For paraphrasing, we use the following:

*"Paraphrase the following: RESPONSE"*.

We skip examples for which Gemini does not return a response.

### A.5 OMITTED EXPERIMENTAL RESULTS

Figure 2 shows the effect of varying the amount of random token corruption on detection pAUC. We observe the same trend as for AUC. Figure 3 plots a histogram of the entropy of the underlying next-token probability distribution under temperature 1 random sampling without watermarking across our dataset. We see the entropy is concentrated between 0.5 and 3 nats. We plot the AUC lower bound predicted by Theorem 4.2 ($k = 1, m = 1024$) sweeping *our* entropy term $\alpha$ across this range, with the understanding that for sufficiently large $m$, our $\alpha$ is a good estimator of the true underlying entropy. In Figure 4 we look at the impact of $m$ and $T$ on our AUC bound when the optimal $\alpha = \log(m)$ is plugged in. We see sharp diminishing returns w.r.t. $m$ (performance saturates after around $m = 10$ for all $T$'s). We empirically observe this saturation in Table 1, where AUC saturated at 97.7% at $m = 16$ — that is, increasing $m$ beyond 16 had negligible impact. Furthermore, we observe that the bound increases sharply with $T$, corroborating the trend we see empirically in Figure 1.

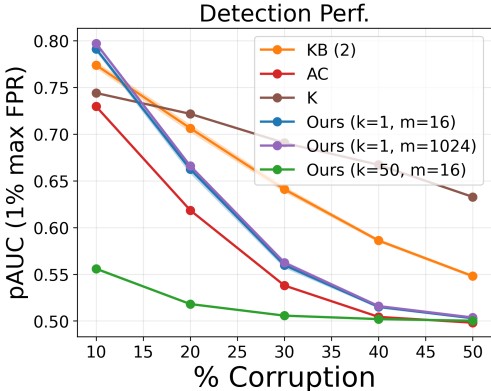

Figure 2: Effect of the amount of (random token replacement) corruption on detection pAUC (mixed $T$'s) with 1% max FPR.

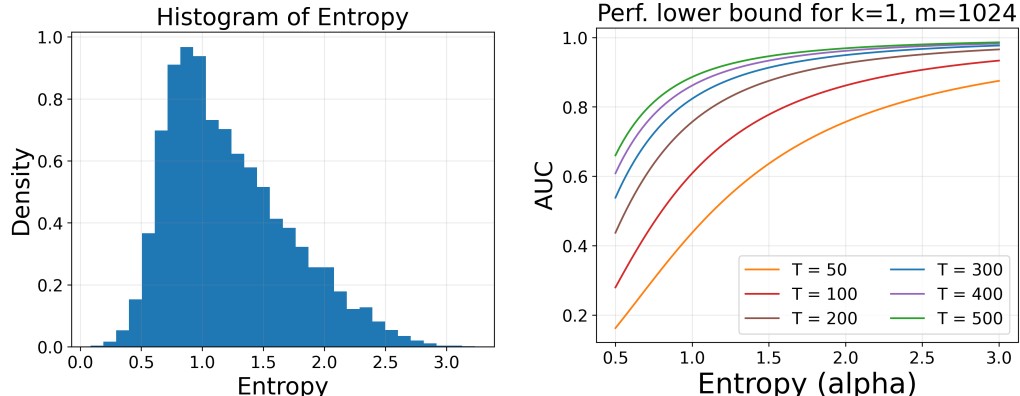

Figure 3: **Left**: Histogram of the average entropy (nats) in the LLM's underlying next-token distribution across non-watermarked response tokens. **Right**: A lower bound for ROC-AUC predicted by Theorem 4.2 as a function of the entropy term $\alpha$ for the range of values we observe empirically. When $m$ is large, $\alpha$ becomes a reasonable estimator of the LLM's entropy.

Given a next-token distribution over the vocabulary, we can estimate $\alpha$ via simulation. In Figure 5 we plot the effect of $m$ on $\hat{\alpha}$, our simulated entropy, for two distributions $p$ — uniform and Zipf — over a 32k token vocabulary. Neither may be realistic in practice, but the exercise is still informative as we observe that $\hat{\alpha}$ follows $\log(m)$ pretty well for even large $m$'s when $p$ is uniform. As expected, $\hat{\alpha}$ is smaller when $p$ is Zipf (lower entropy) and deviates from $\log(m)$ for large $m$.

Figure 7 plots the performance that Theorem 4.4 predicts when using the optimal likelihood ratio test with the Gamma distribution.

Table 6 shows perplexity and detection performance for GEMMA-7B-INSTRUCT on the *eli5-category* dataset. The trends here are as before. Figure 6 shows the impact of number of test samples on detection.

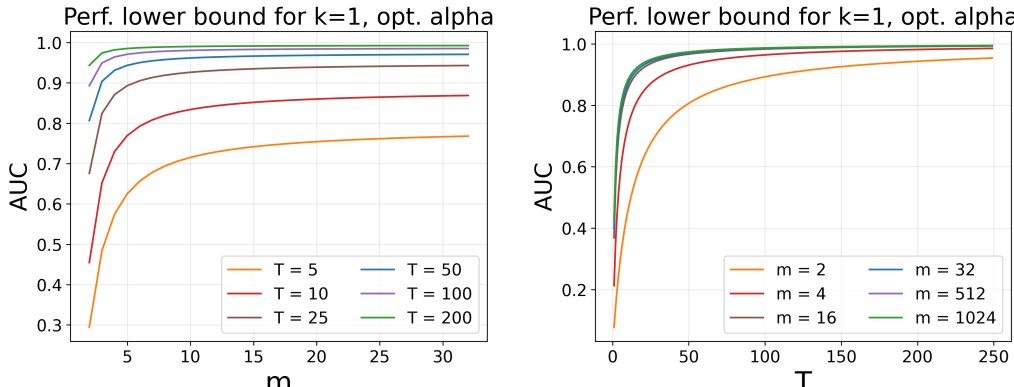

Figure 4: **Left**: A lower bound for ROC-AUC predicted by Theorem 4.2 as a function of $m$ (using optimal $\alpha = \log(m)$). **Right**: Same plot, but as a function of $T$ (again, using optimal $\alpha$).

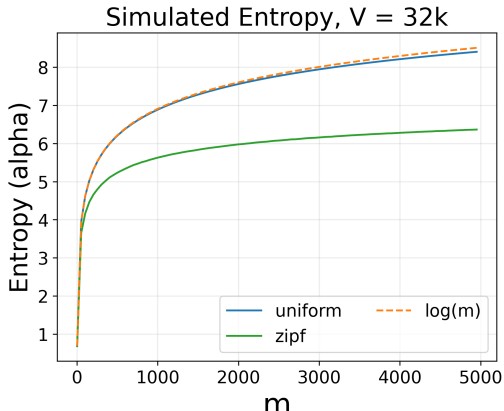

Figure 5: Given a distribution over the vocabulary (taken to be of size 32k), we can estimate $\alpha$ for finite $m$ via simulation (1000 trials). We observe that when the underlying next-token distribution is uniform, $\alpha \approx \log(m)$ in a practical range for $m$. However, when the underlying distribution is Zipf (less entropy), $\alpha$ quickly deviates from $\log(m)$ as $m$ grows and the probability of sampling duplicate tokens increases.

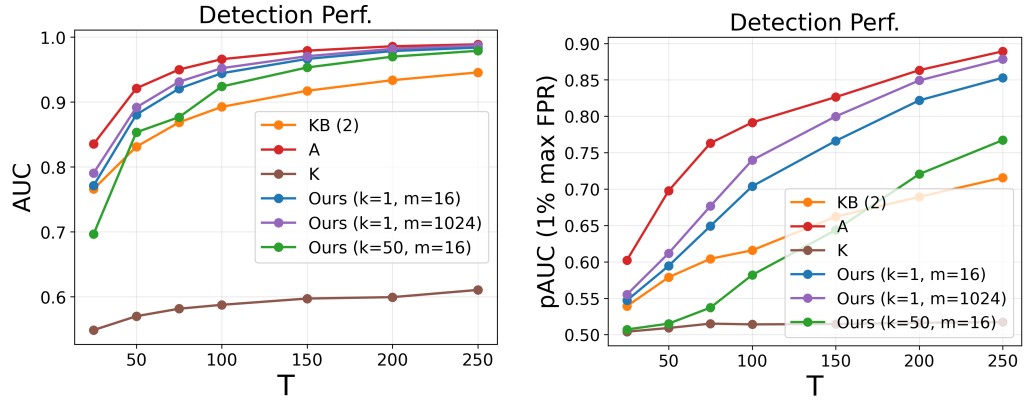

Figure 6: Impact of number of test samples $T$ on detection performance for GEMMA-7B-INSTRUCT on *eli5-category*

| | PPL | LH | AUC | pAUC | C. AUC | C. pAUC |
|---|---|---|---|---|---|---|
| Max Std. Error | 0.03 | 0.002 | 0.1 | 0.1 | 0.2 | 0.1 |
| Unif. Fisher $p$-value | | | | | | |
| Flat ($k = 1$) | | | | | | |
| 2 | 3.46 | 0.597 | 86.3 | 62.3 | 75.8 | 54.7 |
| 4 | 3.36 | 0.604 | 94.8 | 79.0 | 87.6 | 66.4 |
| 16 | 3.20 | 0.618 | 97.9 | 90.0 | 94.0 | 80.2 |
| 32 | 3.06 | 0.629 | 98.2 | 91.7 | 94.9 | 82.7 |
| 512 | 2.63 | 0.668 | 98.5 | 93.0 | 95.7 | 85.5 |
| 1024 | 2.61 | 0.670 | 98.5 | 93.2 | 95.7 | 85.8 |
| Flat ($k = 10$) | | | | | | |
| 2 | 4.10 | 0.568 | 78.9 | 53.1 | 67.4 | 51.0 |
| 4 | 4.06 | 0.572 | 91.2 | 65.3 | 80.5 | 55.0 |
| 16 | 3.86 | 0.583 | 97.1 | 82.8 | 90.9 | 68.1 |
| 32 | 3.80 | 0.587 | 97.9 | 86.3 | 92.7 | 72.7 |
| Flat ($k = 50$) | | | | | | |
| 2 | 3.79 | 0.581 | 65.5 | 50.5 | 57.1 | 50.2 |
| 4 | 3.76 | 0.584 | 78.4 | 52.0 | 66.1 | 50.7 |
| 16 | 3.72 | 0.586 | 89.8 | 60.2 | 77.9 | 53.1 |
| 32 | 3.67 | 0.589 | 92.0 | 64.7 | 80.7 | 54.5 |

Table 2: Results (10% corruption, 1% max FPR) for $U(0, 1)$ when a meta $p$-value is used for scoring, wherein the $T$ $n$-gram-level $p$-values are combined using Fisher's method. The $k = 1$ setting is precisely a stochastic version of Aaronson Corrected. AUCs, pAUCS and their standard errors are scaled by 100.

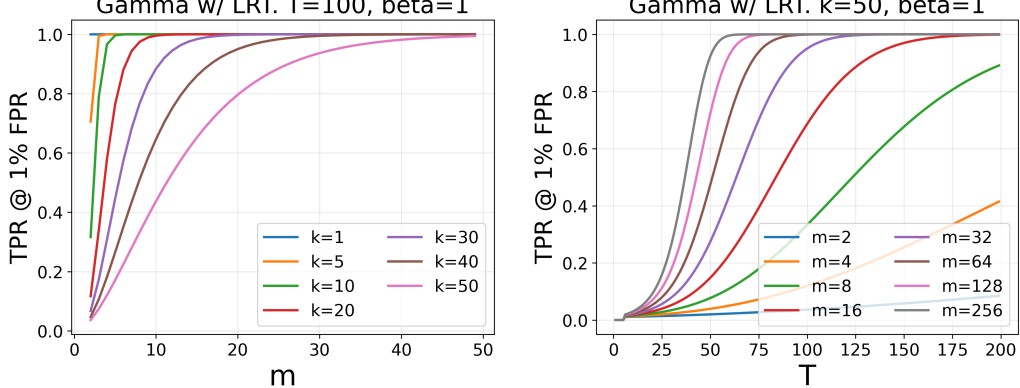

Figure 7: Detection performance (TPR at 1% FPR) of the likelihood ratio test (LRT) predicted by Theorem 4.4. **Left**: Effect of $m$, the number of sampled sequences, for various sequence lengths $k$, when the number of test samples $T = 100$. **Right**: Effect of $T$ for various $m$'s when $k = 50$. We see that degradation due to large $k$ can be offset by using a larger $m$ and that the hit from small $m$ can be compensated by large $T$.

|  |  | PPL | LH | AUC | pAUC | C. AUC | C. pAUC |
|---|---|---|---|---|---|---|---|
| Max Std. Error |  | 0.03 | 0.002 | 0.1 | 0.3 | 0.2 | 0.1 |
| Unif. KDE LRT |  |  |  |  |  |  |  |
| Flat $(k=1)$ | 2 | 3.46 | 0.597 | 78.1 | 57.7 | 64.4 | 51.5 |
|  | 4 | 3.36 | 0.604 | 73.9 | 56.0 | 60.7 | 51.2 |
|  | 16 | 3.20 | 0.618 | 66.6 | 53.9 | 56.7 | 51.3 |
|  | 32 | 3.06 | 0.629 | 64.0 | 53.6 | 55.2 | 51.3 |
|  | 512 | 2.63 | 0.668 | 56.2 | 51.3 | 50.4 | 50.3 |
|  | 1024 | 2.61 | 0.670 | 55.4 | 51.1 | 49.9 | 50.2 |
| Flat $(k=10)$ | 2 | 4.10 | 0.568 | 84.1 | 58.0 | 72.8 | 53.1 |
|  | 4 | 4.06 | 0.572 | 94.8 | 72.9 | 83.9 | 58.7 |
|  | 16 | 3.86 | 0.583 | 97.8 | 85.1 | 88.3 | 64.2 |
|  | 32 | 3.80 | 0.587 | 97.3 | 85.6 | 86.9 | 64.0 |
| Flat $(k=50)$ | 2 | 3.79 | 0.581 | 69.0 | 51.6 | 60.9 | 50.9 |
|  | 4 | 3.76 | 0.584 | 83.1 | 55.6 | 71.0 | 52.4 |
|  | 16 | 3.72 | 0.586 | 94.0 | 68.2 | 81.8 | 56.2 |
|  | 32 | 3.67 | 0.589 | 95.5 | 72.5 | 84.0 | 57.9 |
| Gamma Exact LRT |  |  |  |  |  |  |  |
| Flat $(k=1)$ | 2 | 3.45 | 0.598 | 76.6 | 57.0 | 63.8 | 51.6 |
|  | 4 | 3.44 | 0.600 | 74.4 | 55.2 | 61.5 | 51.2 |
|  | 16 | 3.17 | 0.623 | 68.3 | 53.6 | 57.8 | 51.3 |
|  | 32 | 3.04 | 0.634 | 65.5 | 53.5 | 56.2 | 51.5 |
| Flat $(k=10)$ | 2 | 4.07 | 0.570 | 82.9 | 58.4 | 70.3 | 52.8 |
|  | 4 | 4.01 | 0.573 | 89.4 | 67.5 | 73.4 | 54.1 |
|  | 16 | 3.96 | 0.577 | 85.1 | 61.4 | 68.0 | 51.7 |
|  | 32 | 3.93 | 0.580 | 82.1 | 57.7 | 65.7 | 51.2 |

Table 3: Results when the likelihood-ratio test is used for scoring in place of $p$-values. When $F = U(0, 1)$, the null and alternative likelihoods are estimated non-parametrically using kernel density estimation (KDE). When $F = -\text{Gamma}(1/k, 1)$, the densities given in Theorem 4.4 are used. AUCs, pAUCs, and their standard errors are scaled by 100.

| | PPL | LH | AUC | pAUC | C. AUC | C. pAUC |
|---|---|---|---|---|---|---|
| Max Std. Error | 0.04 | 0.002 | 0.1 | 0.2 | 0.1 | 0.2 |
| $F = N(0,1)$ | | | | | | |
| Flat ($k=1$)  2 | 3.47 | 0.597 | 90.4 | 68.7 | 81.7 | 58.5 |
| 4 | 3.36 | 0.605 | 95.9 | 83.0 | 90.2 | 70.7 |
| 16 | 3.15 | 0.622 | 98.0 | 90.6 | 94.2 | 80.4 |
| 32 | 3.05 | 0.631 | 98.2 | 91.8 | 94.9 | 82.2 |
| 512 | 2.72 | 0.661 | 98.5 | 92.9 | 95.4 | 83.8 |
| 1024 | 2.70 | 0.663 | 98.5 | 93.0 | 95.4 | 84.1 |
| Flat ($k=10$)  2 | 4.13 | 0.567 | 84.1 | 56.3 | 73.3 | 52.1 |
| 4 | 4.02 | 0.573 | 94.2 | 73.3 | 85.8 | 59.8 |
| 16 | 3.93 | 0.579 | 98.0 | 87.9 | 93.2 | 74.5 |
| 32 | 3.84 | 0.584 | 98.4 | 90.0 | 94.1 | 77.7 |
| Flat ($k=50$)  2 | 3.82 | 0.580 | 71.0 | 50.9 | 62.5 | 50.4 |
| 4 | 3.73 | 0.585 | 83.8 | 53.9 | 72.4 | 51.5 |
| 16 | 3.69 | 0.588 | 93.0 | 67.5 | 83.1 | 55.9 |
| 32 | 3.67 | 0.589 | 94.5 | 72.7 | 85.6 | 58.6 |
| $F = \chi_2^2$ | | | | | | |
| Flat ($k=1$)  2 | 3.45 | 0.597 | 86.2 | 62.1 | 75.5 | 54.5 |
| 4 | 3.39 | 0.602 | 94.8 | 79.1 | 87.8 | 66.8 |
| 16 | 3.20 | 0.617 | 97.9 | 90.1 | 93.9 | 80.1 |
| 32 | 3.08 | 0.627 | 98.2 | 91.7 | 94.9 | 82.9 |
| 512 | 2.98 | 0.644 | 98.7 | 95.2 | 96.7 | 89.6 |
| 1024 | 3.03 | 0.641 | 98.8 | 95.7 | 97.0 | 90.5 |
| Flat ($k=10$)  2 | 4.12 | 0.567 | 81.6 | 54.4 | 69.8 | 51.4 |
| 4 | 4.04 | 0.573 | 93.5 | 70.3 | 84.0 | 57.7 |
| 16 | 3.84 | 0.585 | 98.1 | 87.5 | 93.1 | 74.1 |
| 32 | 3.65 | 0.596 | 98.7 | 90.6 | 94.7 | 78.6 |
| Flat ($k=50$)  2 | 3.77 | 0.583 | 68.1 | 50.6 | 58.7 | 50.2 |
| 4 | 3.74 | 0.585 | 82.0 | 52.9 | 69.4 | 51.0 |
| 16 | 3.68 | 0.588 | 92.9 | 65.6 | 81.9 | 55.0 |
| 32 | 3.65 | 0.591 | 94.5 | 71.5 | 84.5 | 57.7 |

Table 4: Results (10% corruption, 1% max FPR) when $F$ is $N(0,1)$ or $\chi_2^2$ and $p$-values are used for scoring. AUCs, pAUCS, and their standard errors are scaled by 100.

| | LH | AUC | pAUC | C. AUC | C. pAUC | P. AUC | P. pAUC |
|---|---|---|---|---|---|---|---|
| Greedy Decoding | 0.814 | - | - | - | - | - | - |
| Random Sampling | 0.593 | - | - | - | - | - | - |
| Aaronson | 0.654 | 71.8 | 67.7 | 65.7 | 62.6 | 52.4 | 50.9 |
| Aaronson Cor. | 0.654 | 98.3 | 92.9 | 95.4 | 84.7 | 57.6 | 51.7 |
| Kirchenbauer 0.5 | 0.596 | 70.7 | 51.7 | 68.3 | 51.2 | 47.3 | 49.9 |
| 1 | 0.594 | 85.4 | 59.9 | 81.9 | 56.5 | 50.9 | 50.0 |
| 2 | 0.569 | 96.6 | 82.5 | 94.8 | 76.5 | 55.7 | 50.4 |
| 3 | 0.522 | 99.1 | 94.0 | 98.4 | 90.4 | 60.5 | 51.3 |
| 4 | 0.493 | 99.8 | 98.2 | 99.6 | 96.6 | 63.9 | 52.4 |
| Kuditipudi | 0.592 | 85.8 | 76.5 | 85.1 | 74.3 | 67.5 | 52.2 |
| Flat (k = 1) 2 | 0.597 | 90.5 | 69.7 | 82.6 | 59.4 | 50.6 | 50.1 |
| 4 | 0.604 | 96.0 | 83.7 | 90.6 | 71.4 | 51.6 | 50.5 |
| 16 | 0.618 | 97.7 | 90.2 | 94.1 | 79.9 | 53.0 | 50.8 |
| 32 | 0.629 | 97.9 | 90.7 | 94.4 | 80.8 | 52.8 | 50.7 |
| 512 | 0.668 | 97.8 | 90.5 | 94.3 | 80.5 | 53.2 | 50.9 |
| 1024 | 0.670 | 97.8 | 90.5 | 94.2 | 80.5 | 52.4 | 50.7 |
| Flat (k = 10) 2 | 0.568 | 84.0 | 56.5 | 74.3 | 52.3 | 49.2 | 50.0 |
| 4 | 0.572 | 94.1 | 73.8 | 86.2 | 60.2 | 51.0 | 50.1 |
| 16 | 0.583 | 97.9 | 87.7 | 93.2 | 74.2 | 53.5 | 50.4 |
| 32 | 0.587 | 98.3 | 89.7 | 94.2 | 77.7 | 54.1 | 50.5 |
| Flat (k = 50) 2 | 0.581 | 70.5 | 50.9 | 63.1 | 50.5 | 47.6 | 50.0 |
| 4 | 0.584 | 83.5 | 54.1 | 72.7 | 51.6 | 49.5 | 50.0 |
| 16 | 0.586 | 93.0 | 67.9 | 83.7 | 56.3 | 50.2 | 50.1 |
| 32 | 0.589 | 94.5 | 72.9 | 86.0 | 59.0 | 51.4 | 50.2 |
| Rec. (k = 1) 4 | 0.601 | 93.9 | 78.2 | 87.3 | 65.8 | 50.0 | 50.3 |
| 16 | 0.607 | 95.4 | 83.5 | 90.8 | 72.5 | 53.5 | 50.7 |
| 32 | 0.612 | 96.5 | 85.8 | 92.0 | 74.5 | 50.4 | 50.6 |
| 512 | 0.632 | 97.4 | 88.6 | 92.9 | 77.5 | 51.0 | 51.1 |
| Rec. (k = 10) 4 | 0.567 | 89.6 | 64.9 | 80.3 | 55.6 | 49.1 | 50.0 |
| 16 | 0.568 | 93.6 | 74.8 | 87.0 | 62.4 | 53.0 | 50.2 |
| 32 | 0.573 | 95.1 | 78.0 | 88.6 | 64.4 | 51.2 | 50.2 |
| Rec. (k = 50) 4 | 0.582 | 75.9 | 52.2 | 67.0 | 51.0 | 48.1 | 50.0 |
| 16 | 0.583 | 81.5 | 55.0 | 73.7 | 52.2 | 52.1 | 50.2 |
| 32 | 0.582 | 84.0 | 56.6 | 75.3 | 52.6 | 49.7 | 50.0 |

Table 5: Average per-token likelihoods and detection performance when the negative class is taken to be non-watermarked generations sampled with temperature 1. The trends here are consistent with those discussed in the main text, where the negative class consists of non-watermarked argmax / greedy generations and perplexity is used to measure distortion. AUCs and pAUCS are scaled by 100.

|  | PPL | LH | AUC | pAUC |
|---|---|---|---|---|
| Greedy Decoding | 1.313 | 0.872 | - | - |
| Random Sampling | 1.627 | 0.811 | - | - |
| Aaronson | 1.619 | 0.814 | 61.0 | 57.8 |
| Aaronson Cor. | 1.619 | 0.814 | 93.0 | 70.9 |
| Kirchenbauer 0.5 | 1.649 | 0.808 | 61.6 | 50.7 |
| 1 | 1.673 | 0.803 | 72.1 | 52.3 |
| 2 | 1.836 | 0.782 | 87.8 | 63.0 |
| 3 | 2.159 | 0.743 | 95.3 | 78.5 |
| 4 | 2.847 | 0.683 | 98.3 | 90.0 |
| Kuditipudi | 1.615 | 0.814 | 58.4 | 51.0 |
| Flat ($k = 1$) 2 | 1.631 | 0.810 | 77.1 | 53.6 |
| 4 | 1.623 | 0.811 | 87.0 | 61.7 |
| 16 | 1.621 | 0.812 | 92.4 | 70.3 |
| 32 | 1.615 | 0.812 | 92.8 | 71.9 |
| 512 | 1.610 | 0.814 | 93.2 | 73.1 |
| 1024 | 1.610 | 0.814 | 93.2 | 72.9 |
| Flat ($k = 10$) 4 | 1.657 | 0.807 | 89.4 | 61.7 |
| 16 | 1.653 | 0.808 | 94.7 | 75.0 |
| Flat ($k = 50$) 4 | 1.652 | 0.808 | 80.5 | 52.6 |
| 16 | 1.645 | 0.810 | 89.7 | 60.4 |
| Rec. ($k = 1$) 4 | 1.623 | 0.813 | 82.1 | 57.0 |
| 16 | 1.621 | 0.812 | 87.5 | 63.0 |
| 32 | 1.630 | 0.810 | 88.1 | 63.9 |
| 512 | 1.615 | 0.815 | 90.0 | 66.7 |
| Rec. ($k = 10$) 4 | 1.665 | 0.805 | 84.0 | 56.2 |
| 16 | 1.662 | 0.806 | 89.6 | 64.4 |
| Rec. ($k = 50$) 4 | 1.664 | 0.806 | 73.2 | 51.2 |
| 16 | 1.653 | 0.808 | 79.4 | 53.5 |

Table 6: Main results (mixed $T$'s for AUC and pAUC where max FPR is 1%) for GEMMA-7B-INSTRUCT on the *eli5-category* test split. AUC and pAUC are scaled by 100. We observe the same trends here as with MISTRAL-7B-INSTRUCT on *databricks-dolly-15k*. When $k = 1$ and $m = 1024$ (white-box setting) we are slightly better in perplexity and detection (sans corruption) than Kuditipudi et al. (2023) and on-par with Aaronson (2023). Kirchenbauer et al. (2023a) can always outperform on detection by cranking up $\delta$, but when matched on perplexity, we achieve better detection. For example, $\delta = 0.5$ gives perplexity of 1.649 and AUC of 61.6% whereas we achieve perplexities / AUC's of 1.610 and 93.2% when $k = 1, m = 1024$ and even 1.645 / 89.7% when $k = 50, m = 16$ (black-box).

|  | AUC | pAUC | C. AUC | C. pAUC | P. AUC | P. pAUC |
|---|---|---|---|---|---|---|
| Aaronson Cor. (sum $p$-value) | 97.1 | 75.2 | 92.5 | 62.9 | 54.6 | 50.0 |

Table 7: Detection performance (mixed $T$'s) when a sum-based $p$-value is used in the length correction of Aaronson (2023). We observe slightly worse performance than using Fisher's method to combine the $p$-values of individual tests. AUCs and pAUCs are scaled by 100.

### A.6 Omitted Proofs

**Lemma A.1.** *Assume all draws from* LM$( \cdot \mid P;\ k)$ *are i.i.d. with distribution $\mu$ and that the unique seeds across $n$-grams and sequences, $\{S_{i,l}\}_{i,l}$ are conditionally independent given the counts of the sampled sequences. Then the output of any number of calls to* WatermarkSingle *with* LM *using key $K$ are also i.i.d. with distribution $\mu$.*

*Proof.* For concreteness, let $\tilde{m}$ be the number of calls to WatermarkSingle, where the $v$-th call draws $m$ samples $\mathbf{Q}_v = \{Q_{(v,1)}, \ldots, Q_{(v,m)}\}$ from LM$( \cdot \mid P;\ k)$. First we show (mutual) independence. We note that because $F$, $m$, $K$, $P$ are all fixed, non-random quantities, the watermark selection process embodied in Algorithm 1 can be seen as a *deterministic* function $\psi_{F,m,K,P}$ that takes $m$ input sequences $\mathbf{Q}_v$ and outputs one of them. The randomness in the deduplication of $n$-grams is a non-issue since it is independent across calls. Since functions of independent random variables are independent and $\{\mathbf{Q}_v\}_{v=1}^{\tilde{m}}$ is independent, so is $\{\psi_{F,m,K,P}(\mathbf{Q}_v)\}_{v=1}^{\tilde{m}}$. This proves independence.

Now, we prove that the outputs are identically distributed with the same distribution as their inputs. To do this, consider the $v$-th call in isolation and for ease of notation, let $\{Q_1, \ldots, Q_m\} = \mathbf{Q}_v$ and $X_w = \psi_{F,n,K,P}(\mathbf{Q}_v)$. Let $\{(X_1, c_1), \ldots, (X_j, c_j)\}$ be the unique sequences and corresponding counts. Note that the $\{(X_i, c_i)\}_i$ need not be independent (it is easy to come up with a counter-example). Let $S_i$ be the integer seeds for $X_i$ after deduplication. Conditioned on $(c_1, \ldots, c_j)$, $\{S_{i,l}\}_{i,l}$ is independent and so $\{R_{i,l}\}_{i,l}$ consists of *i.i.d.* draws from $F$ by virtue of pseudorandomness. As $F$ is also continuous, we have that when conditioned on $(c_1, \ldots, c_j)$, $u_i \overset{iid}{\sim} U(0,1)$ for $i = 1, \ldots, j$, by the inverse-sampling theorem.

Let $x$ be any sequence. We wish to show that $\mathbb{P}(X_w = x) = \mu(x)$. Let $c = \sum_i \mathbf{1}[Q_i = x]$. The independence of the $\mathbf{1}[Q_i = x]$'s follows from the independence of the $Q_i$'s, and thus $c \sim$ Binomial$(m, \mu(x))$. Clearly, $\mathbb{P}(\{x\text{ selected}\} \mid c = 0) = 0$. If $c > 0$ then obviously one of the $X_i$'s is $x$, and we can, without loss of generality, label $X_1 = x$ and $c_1 = c$, so that $\mathbb{P}(\{x\text{ selected}\} \mid c = i) = \mathbb{P}(\{X_1\text{ selected}\} \mid c_1 = i)$. Now,

$$\mathbb{P}(\{X_1\text{ selected}\} \mid c_1, \ldots, c_j) = \mathbb{P}\left(\left\{1 = \text{argmax}_t\, u_t^{m/c_t}\right\} \,\middle|\, c_1, \ldots, c_j\right)$$

$$= \mathbb{P}\left(\left\{1 = \text{argmax}_t\, \frac{\log(u_t)}{c_t/m}\right\} \,\middle|\, c_1, \ldots, c_j\right)$$

$$= \mathbb{P}\left(\{1 = \text{argmin}_t\, \log(-\log(u_t)) - \log(c_t/m)\} \mid c_1, \ldots, c_j\right)$$

$$= \mathbb{P}\left(\{1 = \text{argmax}_t -\log(-\log(u_t)) + \log(c_t/m)\} \mid c_1, \ldots, c_j\right).$$

Let $g_t = -\log(-\log(u_t))$. It is a known fact that if $u_t \overset{iid}{\sim} U(0,1)$, then $g_t \overset{iid}{\sim}$ Gumbel$(0,1)$. Now we can apply what is often referred to the "Gumbel-Max trick" in machine learning. Conditioned on $(c_1, \ldots, c_j)$,

$$\text{argmax}_t\, g_t + \log(c_t/m) \sim \text{Categorial}\left(\frac{c_t/m}{\sum_t c_t/m}\right)_t = \text{Categorial}\left(c_t/m\right)_t.$$

Thus,

$$\mathbb{P}(\{X_1\text{ selected}\} \mid c_1 = i) = \sum_{c_2, \ldots, c_j} \frac{\mathbb{P}(\{X_1\text{ selected}\} \mid c_1 = i, c_2, \ldots, c_j)\mathbb{P}(c_1 = i, c_2, \ldots, c_j)}{\mathbb{P}(c_1 = i)}$$

$$= \frac{i/m\ \mathbb{P}(c_1 = i)}{\mathbb{P}(c_1 = i)} = i/m.$$

Putting it all together, we have that

$$\mathbb{P}(X_w = x) = \sum_{i=0}^{m} \mathbb{P}(\{x\text{ selected}\} \mid c = i)\mathbb{P}(c = i)$$

$$= \sum_{i=0}^{m} \frac{i}{m}\binom{m}{i}\mu(x)^i(1 - \mu(x))^{m-i}$$

$$= \frac{1}{m}m\mu(x) = \mu(x).$$

We have shown that the outputs of WATERMARKSINGLE are mutually independent and carry the same distribution $\mu$ as their inputs. □

**Remark.** *The proof of Lemma A.1 treats the secret key $K$ as fixed (possibly unknown); treating it as random changes the story, as we illustrate with the following toy example.*

Suppose that regardless of the conditioning prompt, the LM outputs one of two sequences — $x_1$ or $x_2$ with equal probability. Let $u_i = \text{SCORESEQS}(F, (x_i), K, n, P)$ for $i \in \{1, 2\}$. If $m$ is very large, then it becomes very likely that $X_1 = x_1$, $X_2 = x_2$ (modulo the labeling) and $c_1 \approx c_2 \approx m/2$ and so $\text{argmax}_{i=1}^2 u_i^{m/c_i} \approx \text{argmax}_i u_i$. The outputs to two sequential calls to WATERMARKSINGLE should not be independent, because the output and key are dependent and the key is shared across calls. Concretely, if the output to the first call is $x_1$ we learn that our scheme with key $K$ prefers $x_1$ over $x_2$, and so we will likely output $x_1$ in the second call. In contrast, if we had not observed the first call (and our prior on the key had not been updated), we may have returned each sequence with equal probability.

*Proof of Theorem 4.1.* We first show that WATERMARKSINGLE and WATERMARKRECURSIVE are distortion-free and then that autoregressive calls to them as done by WATERMARK preserves this property.

To show WATERMARKSINGLE is distortion-free, we observe that the LM argument supplied is the true underlying language model $\mu$ and that our stochastic samples from the model are i.i.d., so we can apply Lemma A.1 directly.

Distortion-free for WATERMARKRECURSIVE follows easily from induction on $t$, the number of keys (and hence the number of recursive calls). When $t = 1$, the LM is the true underlying language model, so the outputs are i.i.d. from $\mu$. We get $t = v + 1$ by combining Lemma A.1 with the inductive step — that the outputs of WATERMARKRECURSIVE with keys $(K_2, \ldots, K_{v+1})$ are i.i.d. from $\mu$.

Finally, we show that autoregressive decoding where sequences no longer than $k$ tokens are generated one at a time via watermarking continues to be distortion-free.

To do this, we introduce two sets of random variables: $\{X_u^{(i)}\}_{i=1}^\infty$ represents $k$-sized chunks of the model's response when watermarking is *not* employed — that is, $X_u^{(i)}$ represents non-watermarked response tokens for indices $(i-1)k + 1$ to $ik$. Unused chunks can be set to a sentinel value like $\phi$. $\{X_w^{(i)}\}_i$ represents the same collection but when WATERMARK is employed. Let $x$ be a sequence of any length. Partition $x$ into contiguous $k$-sized chunks $(x_1, \ldots, x_t)$. Note that $x_t$ may have length less than $k$ if the stop-token was reached in that chunk, but all other chunks have exactly $k$ tokens. With $P$ as the original prompt, we need to show $\mathbb{P}(X_w = x \mid P) = \mathbb{P}(X_u = x \mid P)$, where $X_w$ and $X_u$ are the watermarked and non-watermarked responses of any length.

$$\mathbb{P}(X_w = x \mid P) = \mathbb{P}(X_w^{(t)} = x_t \mid X_w^{(t-1)} = x_{t-1}, \ldots, X_w^{(1)} = x_1, P) \cdots \mathbb{P}(X_w^{(1)} = x_1 \mid P)$$
$$= \mathbb{P}(X_w^{(1)} = x_t \mid (P, x_1, \ldots, x_{t-1})) \cdots \mathbb{P}(X_w^{(1)} = x_1 \mid P)$$

Because WATERMARKSINGLE and WATERMARKRECURSIVE are distortion-free:

$$= \mathbb{P}(X_u^{(1)} = x_t \mid (P, x_1, \ldots, x_{t-1})) \cdots \mathbb{P}(X_u^{(1)} = x_1 \mid P)$$
$$= \mathbb{P}(X_u = x).$$

□

*Proof of Theorem 4.3.* First consider the flat scheme. Under the null, given our assumption of independence, $R_j \overset{iid}{\sim} F$, so $F_{|R|}\left(\sum_j R_j\right) \sim U(0, 1)$ and the result follows. For the recursive scheme, we know from the flat scheme and from assumed independence that $P_j \overset{iid}{\sim} U(0, 1)$, where $P_j$ is the $p$-value associated with the $j$-th key. Thus, $y \sim \chi^2_{2|P|}$ so that $\chi^2_{2|P|}(y) \sim U(0, 1)$. □

**Lemma A.2.** *Assume the conditions of Theorem 4.2. Conditioned on the counts $c$ of each token in the vocabulary, and which token id $i^*$ was selected (i.e. is the argmax), $u_{i^*} \sim Beta(m/c_{i^*}, 1)$.*

*Proof of Lemma A.2.* Let $z_i = -m\log(u_i)/c_i$, where $u_i \overset{iid}{\sim} U(0,1)$. Then, $z_i \sim \text{Exp}(c_i/m)$ and

$$i^* = \text{argmax}_{i=1}^{j} u_i^{m/c_i} = \text{argmin}_i -m\log(u_i)/c_i = \text{argmin}_i z_i.$$

By nice properties of the Exponential, we have that

$$z_{i^*} \sim \text{Exp}\left(\sum_i \frac{c_i}{m}\right) = \text{Exp}(1).$$

$u_{i^*} = \exp(-c_{i^*}z_{i^*}/m)$, so

$$\mathbb{P}(u_{i^*} \leq t) = \mathbb{P}(z_{i^*} \geq -m\log(t)/c_{i^*}) = \exp(m\log(t)/c_{i^*}) = t^{m/c_{i^*}}.$$

Differentiating this w.r.t to $t$, we recover the pdf of $\text{Beta}(m/c_{i^*}, 1)$. $\qquad\qquad\square$

*Proof of Theorem 4.2.* $F$ is $U(0,1)$. The detection score is $F_T\left(\sum_j R_j\right)$ with $R_j \overset{iid}{\sim} F$ under $\mathcal{H}_0$ and when conditioned on the counts $C$ and the argmax token ids $I^*$, $R_j \sim \text{Beta}\left(m/C_{j,I_j^*}, 1\right)$ under $\mathcal{H}_1$. Redefine $s_0$ and $s_1$ to be $\sum_j R_j$ under $\mathcal{H}_0$ and $\mathcal{H}_1$ respectively.

$$\mathbb{P}(F_T(s_1) \geq F_T(s_0)) = \mathbb{P}(s_1 \geq s_0) = \mathbb{E}_t(s_1 \geq t),$$

where $t \sim \text{IrwinHall}(T)$ since $s_0$ is the sum of $T$ i.i.d. $U(0,1)$'s. Our task now is to find a lower-bound for $s_1$. Noting independence across tokens and that $R_j \in [0,1]$, we can use Popoviciu's bound on variance to obtain,

$$\mathbb{V}(s_1) = \sum_j \mathbb{V}(R_j) \leq \frac{T}{4}(1-0)^2 = T/4.$$

Plugging in the expectation of a Beta and recalling that when conditioned on $C$, the probability that token $i$ in the vocabulary is the argmax token at step $j$ is $C_{j,i}/m$, we have

$$\mathbb{E}(s_1) = \sum_{j=1}^{T} \mathbb{E}_C \left(\sum_{i=1}^{V} \frac{C_{j,i}/m}{1 + C_{j,i}/m}\right).$$

With tedious calculation, it can be shown that

$$\frac{x}{1+x} \geq \frac{x}{2} - \lambda x \log(x), \text{ for } x = \frac{j}{m}, j \in [1,\dots,m], \text{ where}$$

$$\lambda = \frac{1}{\log(m)}\left(\frac{m}{m+1} - \frac{1}{2}\right).$$

Thus,

$$\mathbb{E}(s_1) \geq \sum_{j=1}^{T}\left(\frac{1}{2} - \lambda\mathbb{E}_C \sum_{i=1}^{V} \mathbf{1}\left[C_{j,i} > 0\right] \frac{C_{j,i}}{m}\log\left(\frac{C_{j,i}}{m}\right)\right).$$

$$= \sum_j 1/2 + \lambda\alpha = T/2 + \lambda T\alpha.$$

With bounds on expectation and variance, we proceed to upper-bound the error. Firstly, we have that,

$$\mathbb{E}(s_1 - s_0) \geq T/2 + \lambda T\alpha - T/2 = \lambda T\alpha \geq 0,$$
$$\mathbb{V}(s_1 - s_0) \leq T/4 + T/12 = T/3.$$

$$\mathbb{P}(s_1 \leq s_0) = \mathbb{P}(s_1 - s_0 - \mathbb{E}(s_1 - s_0) \leq -\mathbb{E}(s_1 - s_0))$$
$$\leq \mathbb{P}(s_1 - s_0 - \mathbb{E}(s_1 - s_0) \leq -\lambda T\alpha)$$
$$\leq \frac{\mathbb{V}(s_1 - s_0)}{\mathbb{V}(s_1 - s_0) + (\lambda T\alpha)^2}$$
$$\leq \frac{1}{1 + 3T\lambda^2\alpha^2},$$

where the penultimate line follows from Cantelli's inequality. Thus, we have that

$$\mathbb{P}(s_1 \geq s_0) = 1 - \mathbb{P}(s_1 \leq s_0) \geq \frac{1}{1 + 1/(3T\lambda^2\alpha^2)}.$$

$\square$

*Proof of Theorem 4.4.* Let $r$ be the PRF value for some $n$-gram from the text we wish to text. Let $F_0 = -\text{Gamma}(1/k, \beta)$ with pdf $f_0$ and $F_1 = -\text{Gamma}(1/k, m\beta)$ with pdf $f_1$. By definition, $r \sim F_0$ under $\mathcal{H}_0$. By our assumptions, $c_i = 1$ and $|R_i| = k$, $\forall i$. So, $\text{argmax}_{i=1}^m u_i^{m/c_i} = \text{argmax}_{i=1}^m u_i = \text{argmax}_i F_k\left(\sum_j R_{i,j}\right) = \text{argmax}_i \sum_j R_{i,j} = \text{argmin}_i - \sum_j R_{i,j}$, where the second-to-last equality follows from the monotonicity of $F_k$. $-\sum_j R_{i,j} \sim \text{Gamma}(k/k, \beta) = \text{Exp}(1, \beta)$. $\sum_j R_{i^*,j} \sim -\text{Exp}(1, m\beta)$, because the minimum of Exponentials is Exponential. Thus, $\forall j$, $R_{i^*,j} \sim -\text{Gamma}(1/k, m\beta) = F_1$ and $r \sim F_1$ under $\mathcal{H}_1$. Now let $R$ refer to the $T$ test-time PRF values. From the independence of test $n$-grams, the log-likelihood ratio test has score $s(R) = \sum_{i=1}^T (\log f_1(R_i) - \log f_0(R_i))$ and the fact that it is the uniformly most powerful test follows directly from the Neyman–Pearson lemma. We now have that,

$$f_0(r) = \frac{\beta^{1/k}}{\Gamma(1/k)}(-r)^{1/k-1}\exp(\beta r),$$

$$f_1(r) = \frac{m^{1/k}\beta^{1/k}}{\Gamma(1/k)}(-r)^{1/k-1}\exp(m\beta r),$$

$$s(R) = \frac{T}{k}\log(m) + (m-1)\beta\sum_{i=1}^T R_i, \text{ so that}$$

$$P_{\mathcal{H}_0}(s > t) = P_{\mathcal{H}_0}\left((m-1)\beta\sum_i R_i > t - \frac{T}{k}\log(m)\right) = \text{Gamma}(T/k, \beta)(Q(t)), \text{ and}$$

$$P_{\mathcal{H}_1}(s \leq t) = P_{\mathcal{H}_1}\left((m-1)\beta\sum_i R_i \leq t - \frac{T}{k}\log(m)\right) = 1 - \text{Gamma}(T/k, m\beta)(Q(t)), \text{ where}$$

$$Q(t) = \frac{T\log(m)/k - t}{(m-1)\beta}.$$

$\square$

