# OpenReview forum: "A Watermark for Black-Box Language Models"
_ICLR.cc/2025/Conference — Submitted to ICLR 2025_

### Official Review · Reviewer_Lnyz · 2024-10-20

**Soundness:** 2
**Presentation:** 2
**Contribution:** 2
**Rating:** 5
**Confidence:** 4

**Summary:**

The paper presents watermarking schemes for LLM’s outputs, in the setting that we only have black-box access to the model’s “next token generation” function.

They claim their scheme is “distortion free” and “can be used in a nested way”.

In a bit more detail, the paper’s scheme is based on a scoring function, which in turn is based on a secret key. Then, when the LLM’s output is being generated, at each step, multiple samples are gathered. Then, the scoring function is applied to them all and the one with the highest score is chosen.

*post rebuttal comment*

After the interaction with author(s), I added several comments to the discussion board explaining why I increased my score. I thought I'd add them to this summary as well, just in case the authors might find the comments (hopefully) useful as well. The comments follow:

I thought I would go over the points discussed and say what my final thoughts are, and why (despite remaining disagreements basically about all the points being discussed) I am indeed happier with the paper now and will increase my score. Also, it took me a bit of time to have a closer look at the paper by Christ et al and also come back to the paper to understand their contribution in light of the discussions and what is done in Christ et al.

To me, the main downside of the paper is actually its writing. It is hard to understand their scheme (with such dense descriptions) and why it is objectively and concretely better than previous work. In particular, key concepts need to be formally defined and discussed. The assumption of the paper about independence of the hash of the n-grams (called assumption (**) below). This needs to be *mathematically* and formally written and analyzed. Another major issue is their notion of “soundness”. They show that the detection algorithm can detect watermarked text from honestly generated non-watermarked text (of the same mode). In comparison, Christ et al. show that the soundness holds for any string that is generated independent of the secret key, which I think is the right definition of soundness. In fact, the paper should have a clear soundness definition to begin with. There is a chance that the authors can address this in their final/next draft well by expanding their theoretical claim of Theorem 4.2. Also note that this is where theory is needed, as no experiment can prove robustness for all strings generated independently of the key.

Having said the above, the reason that I am happier with the paper are:

I think the assumption (**) could be proved true if we model the hash function as a random oracle (which is a standard model called the random oracle model ROM in cryptography and allows proving heuristic assumptions in a meaningful way). This, however, is something that authors need to check and argue about, as I am not fully confident about it based on the written material. If one can justify their assumption (**) (eg., in the ROM model), then the entropy assumption (that is provably needed) is not affecting the distortion-freeness of the paper’s scheme (and is only needed for arguing soundness) while in Christ et al, it affects both distortion freeness and soundness. So, this is an interesting aspect that could be a selling point. The authors said that their scheme can also detect strings that only have a common substring (of high entropy) with the original generated text. This would match that of Christ et al, and it is in fact a form of robustness guarantee (though limited).

Now, some comments and responses to the points raised during the discussion with authors:

The authors unfortunately keep saying that Chrsit et al is non-black-box, while my point is that *if you assume* the entropies are large enough (say in every block of 100 tokens) then all of their schemes (not just the 1st one) become fully black-box, as the only thing they need is to accumulate entropies. So, this is where I had trouble evaluating their contribution of being the first black-box scheme in comparison with that of Christ et al, because both schemes could be black-box *based on an assumption*.

The authors confirm that their own scheme can also be “substring-complete” (SC) which is great, but then they bring up Section 6 of Christ et al, which is not quite relevant. That section is about removing the watermark under specific settings, but of course those attacks would not contradict their own SC property (which is a robustness guarantee). To understand SC completeness (which there is a disagreement with authors), and why it can be interpreted as a robustness guarantee, please read Def 8 of the archive version of the Christ et al paper, in which the detection algorithm is run only on the substrings of high entropy. So, if one of such substrings survives after adversary’s edits and stays intact in the final string, it can be detected by checking all possible (contiguous) substrings of the final string for detection (there are at most k^2 of them for strings of length k, which is fine). I might be wrong here, but this is not a main point of discussion anyway.

When discussing how to remove the entropy assumption from the first scheme of Christ et al, authors say “Firstly, the while loop may never terminate. [...] And even if it does terminate, it could take an astronomical number of samples before the condition is met”

The assumption on the entropy can be used to show that (with overwhelming probability) the chance of getting 0 is not “too small” (in particular, it can be lower bounded by 1/poly) and then using Chernoff/Hoeffding bounds, one can show that with polynomial samples, the chance of not hitting zero is exponentially small. So, things can be proved to be fine. But this is not a major point of discussion, because even the main (2nd) algorithm of Christ et al also becomes fully black-box if you assume the entropy is large in each block (of say 100 tokens).

Anyway, I think the paper has a lot of potential, but I think it would benefit quite a bit from a major improvement in the presentation of the ideas, clarification of the assumptions, and comparison with previous work.

**Strengths:**

The problem of black-box watermarking is an important problem, and having new schemes in this direction would be interesting. However, as I explain below, the schemes should be clear in what they offer and what is their advantage over previous work.

post rebuttal: I understood some aspects of the paper better and am increasing my score by one unit. I still have concerns (post discussions) about the writing and assumptions of the paper that I will add to the review.

**Weaknesses:**

The main weakness of the paper is that it is barely readable, when one actually wants to understand the scheme and the arguments. The presentation of the scheme is super dense and lacks formality. Instead of introducing ideas one by one, they are jammed and one gets no intuition as to what is goin on, beyond the high level description of “using scores”.

In fact, the paper’s main setting (which seems to be the main novelty) is already used in previous work published in learning venues. For example, this (cited work) from more than year ago (published in COLT) https://eprint.iacr.org/2023/763 exactly studies the setting that the paper does: black-box access to the token generation function, and does use a similar idea of using a hash function to pick the next token by rejecting some. It is also provably robust (under certain conditions) as opposed to the weaker model studied here (random substitution) and comes with clear theorems that prove undetectability (which implies distortion free-nes and utility both).

One main comment for improving the writing:

- Try to define everything formally and at the right pace.
- There are also issues with using crypto terms without clarity. For example, F is a CDF, and then F[s] is a “single draw from a pseudorandom number generator for F seeded by integer seed s” .  I know cryptography well, but I have no idea what this sentence means. Then, it is assumed that F[h(K,w)] is a PRF. What is the citation that this is a PRF whenever F is PRG? (I don’t think this is true actually).
- What is the role of n-gram, l-gram, and their relation with tokens. Sentences like “where we allow the left endpoint to spill over only to…” are super informal and cannot be formally understood and checked.
Theorem 4.1 : what is F, and why should it be continuous? When it comes to efficient algorithms none are actually continuous (everything is discrete) so this is a strange assumption to make.

**Questions:**

My main question is about the novelty of the paper’s setting and its final results. As mentioned above, the work Christ et al already presents provably secure distortion free black-box watermark that is also robust to adversarial attacks (under a formal definition). Can you compare your work with them (and perhaps other similar previous works using crypto and rejection sampling) and explain what exactly the set of features that your work adds?

---

> ### Author Response · Authors · 2024-11-21
>
> Thank you for the review.
>
>
> **RE: "The presentation lacks formality. Instead of introducing ideas one by one..."**
>
> Ideas are formally introduced one by one. We agree it can be on the terse side (and the Algorithm block helps with this, though it had to be put in the Appendix because of space constraints). This is because we have a lot of key content that should be in the main text. Another reviewer raised a similar concern so we will make a bit more space by being clever about where we compress and moving a theorem to the Appendix, and expand the description along with motivation a bit more.
>
> **RE: your reference**
>
> I'm curious, did you actually read that paper? If so, you will have observed that Algorithm 3 on pg.16 is not black box because it requires computing log-probabilities under the model. The same is true with Algorithm 1, which needs to compute entropy under the model. While the paper may be interesting to the cryptographic community, it has no experiments or evals whatsoever which limits its impact in the machine learning community.
>
> **RE: "It is provably robust as opposed to the weaker model studied here"**
>
> Can you point me to the line in their text that says their proposed method is provably more robust to the one we propose here?
>
> **RE: "issues with crypto terms...I have no idea what this sentence means"**
>
> You can imagine my confusion if you struggle to grasp this but you confidently cite the works of Christ et. al.
> If you're also familiar with numpy and python, this would look roughly like:
>
> ```python
> import numpy as np
> x = (1, 2, 3, 4) # some ngram
> pk = 'notsoprivateeh' # private key
> seed = abs(hash(str(x) + pk))
> rng = np.random.default_rng(seed)
> u = rng.uniform()
> ```
>
>
> In practice we use the hashlib library (hashlib.sha256() ) to be more correct.
> I think what could make the paper better is adding a reference implementation in python in the Appendix, which we can easily do for the camera ready.
>
> **RE: "PRF, PNG, etc"**
>
> The goal of the paper is to introduce an effective practical algorithm and prove practically useful theoretical statements about it while being formally correct. We do not want to obfuscate it or get bogged down in esoteric cryptography jargon. With that said, we will clarify and elaborate more on this point in the main text.
>
> **RE: "n-grams" and "l-grams" and "super informal and cannot be formally understood or checked"**
>
> The n-grams are on the tokens, as we clearly state in the text. We also clearly state "l-grams are taken instead for boundary indices with only l-1<n-1 eligible tokens strictly left of it." Let's walk through an example together.
> Suppose the prompt is: "How good is this review?" and the sampled response you wish to score is: "This is a very low quality review.", and the tokenization is at the word level resulting in token ids (1, 2, 3, 4, 5, 6, 7), and the n in n-gram is set to 4, then what our description is saying is that: the basis for computing the score for token 1 is the l-gram (1,). Why? Because this token has 0 tokens strictly left of it, so l-1=0 implies l=1 (add 1 to both sides of the equation).
> Overall, I think the formal description and the algorithm is prescriptive enough for anyone to implement the scheme exactly. With that said, there is no harm in adding a code implementation, which we will do.
>
> **RE: "What is F" in theorem 4.1. "Why does it need to be continuous"**
>
> We clearly state "F is a continuous distribution". We repeatedly mention across the text that F is a CDF: e.g. "If F is a cumulative distribution function (CDF)...".
> At the end of the day we need to obtain a U(0, 1) RV from the token-level scores, which we do by applying $F_k$ (also continuous). From basic probability, we know that if X is a continuous RV with CDF F, then F(X) is U(0, 1) (more here: https://en.wikipedia.org/wiki/Inverse_transform_sampling).
>
> **RE: "when it comes to efficient algorithms, none are actually continuous (everything is discrete), so this is a strange assumption to make"**
>
> If I understand your logic correctly, then it would be quite strange to mention real numbers and computers in the same sentence because a 32 bit floating point number is discrete and numbers are continuous. If you mean that the CDF F is hard to evaluate when F is continuous, what about when F = U(0, 1)? Then F(x) = x. Do you think this is efficient to evaluate? Please correct or qualify your statement more.
>
> **RE: "our contribution compared to Christ et. al"**
>
> Hmm, well, our scheme is truly black-box, has practical theoretical guarantees, and most importantly was evaluated and benchmarked against state-of-the-art baselines on a couple of the best 7B models on datasets representative of real world usage and we include a slew of ablations to understand the mechanisms and tradeoffs at play.
> How does that sound?

---

> > ### Comment · Reviewer_Lnyz · 2024-11-26
> > **Re:**
> >
> > I don’t think the tone of the discussions is heading in the right direction. A paper’s appeal to the crypto community (while it is published in COLT) is not a downside for the learning community. And I certainly don’t find the “esoteric cryptography jargon” reference constructive. I invite the authors to be more calm and focused on the discussion using a respectful tone. I certainly did not mean to demean the paper's contributions. For each of the items of the discussion, for simplicity, I will only quote the short phrase you used and avoid further texts.
> >
> > > RE: "The presentation lacks formality. Instead of introducing ideas one by one..."
> >
> > It is great that you plan to make the paper more readable by being more careful about how to explain ideas in a more accessible way.
> >
> > > RE: your reference
> >
> > The schemes of that paper are actually black-box, if you assume the entropy is not too small in the output of the model. They first describe their (what you below call truly black-box) scheme based on this assumption, and when they want to remove this (actually very natural and even necessary) assumption they describe a scheme that accumulates the entropies, but even in this case they do not need to model parameters, and only use the model’s next-token distributions as black-box (which determines the entropy too). Also, note that if there is not enough entropy in the model’s outputs the watermarking becomes meaningless, as the sentence is predictable information theoretically. (see the sentence “we show that it is inherent that …” in their Section 2.6 about the necessity of entropy. I only brought up this reference, because you make the black-box-ness of your scheme a main focus of your paper (mentioning this in the title and abstract) and I was struggling to appreciate how novel this aspect actually is.
> >
> >
> > > RE: "It is provably robust as opposed to the weaker model studied here"
> >
> > They have theorems about what their robustness guarantee is (look for the term “substring-complete”), and in particular they allow adversaries that can do everything while keeping a sufficiently large sub-string untouched. You also have adversaries, but as far as I understand, yours only uses “random substitution” rather than “worst case substitution” and again as far as I understand you do not have a theorem to prove your robustness. In fact, in light of the recent developments (such as the works of Christ et al) it is more important to address robustness using proofs rather than experiments, as there could always be a next adversary that does not fit into the experiment.
> >
> > > RE: "issues with crypto terms...I have no idea what this sentence means"
> >
> > Thanks for sharing the code. But please note that I did not claim that you don’t know how to implement your algorithms. What I respectfully complained about was the English description of it in the paper, which is a necessary thing to have and I did not find it clear.
> >
> > > RE: "PRF, PNG, etc"
> >
> > Thanks for planning to work on improving the presentation of the paper.
> >
> > > RE: "when it comes to efficient algorithms, none are actually continuous (everything is discrete), so this is a strange assumption to make"
> >
> > What I was trying to say was that when the only thing you need from a distribution is to be continuous, this is something that looks unusual. In particular, it is true that all the distributions we actually work with are discrete. Yet, you are right that we use things like “uniform distribution over [0,1]” abundantly, and they are sometimes very useful, but my question was about a case in which the only thing needed was that it is continuous. Having said that, you clarified the reason here more; thanks.
> >
> > > RE: "our contribution compared to Christ et. al"
> >
> > Having experiments is a plus, but when a major selling point is being black-box, one needs to have a better comparison with previous work that is black-box and discuss the exact details that make one work (here yours) more black-box.

---

> > > ### Author Response · Authors · 2024-11-26
> > >
> > > I apologize for the tone, which may come off as a bit provocative.
> > > Christ et. al is a wonderful paper and that it was published in COLT corroborates this. What I'm saying is that the paper does not have experiments so there are no empirical grounds to believe the scheme works in practice.
> > >
> > > **RE: "the schemes of that paper are actually black-box"**
> > >
> > > This is, respectfully, not true.
> > > You say "even in this case they do not need [the] model parameters, and only use the model's next-token distributions as black-box".
> > > How would you obtain the next-token probabilities without access to the model weights / you are not the model owner? The main LLM providers do not offer this in their APIs. This is the first thing we discuss in our introduction section, defining "white-box" schemes as those that need the next-token logits / probabilities:
> > >
> > > *Furthermore, the detecting party may or may not have white-box access (e.g. an ability to compute log-probabilities) to the generator they wish to test against. Typically, parties that have white-box access are the owners of the model so we refer to this case as first-party detection and the counterpart as third-party detection...Most proposed techniques do not modify the underlying LLM’s model weights or its training procedure but rather inject the watermark during autoregressive decoding at inference time. They require access to the next-token logits and inject the watermark every step of the sampling loop. This required access prevents third-party users of an LLM from applying their own watermark as proprietary APIs currently do not support this option. Supporting this functionality presents a security risk in addition to significant engineering considerations. Concretely, Carlini et al. (2024) showed that parts of a production language model can be stolen from API access that exposes logits. In this work, we propose a watermarking scheme that gives power back to the people — third-party users can watermark a language model given nothing more than the ability to sample sequences from it. Our scheme is faithful to the underlying language model and it can outperform existing white-box schemes.*

---

> > > ### Author Response · Authors · 2024-11-26
> > >
> > > **RE: "also note that if there is not enough entropy in the model's outputs the watermarking becomes meaningless..'in their Section 2.6 about the necessity of entropy'"**
> > >
> > > You are correct that entropy is necessary for effective watermarking. This is common knowledge to anyone versed in the literature. We describe the role of entropy in our scheme at great lengths and quantify the performance of the algorithm directly in **terms of entropy** in Theorem 4.2.
> > >
> > > *Theorem 4.2 connects detection performance to the language model's underlying distribution, number of sampled tokens $m$, and number of test samples $T$. More entropy and more test samples guarantee higher performance. When the model is extremely confident, $\alpha \to 0$ and so does our lower bound. Note that because $\alpha$ measures the entropy of the empirical distribution arising from \emph{sampling} tokens, it depends on both the underlying next-token probability distribution as well as $m$. Concretely, when conditioned on the next-token probabilities $p$, $c \sim \text{Multinomial}\left(m, p\right)$. The largest $\alpha$ is achieved when the $c_i$'s are 1, which can occur when the underlying distribution is uniform (maximal uncertainty) and/or $m$ is not large. In this case, $\alpha \to \log(m)$ and our bound goes to $1/\left(1 + 1/\left(3T\left(\frac{m}{m+1} - \frac{1}{2}\right)^2\right)\right)$. This quantity has very sharp diminishing returns with respect to $m$, so there may be little value in increasing $m$ beyond a certain point. When $m \to \infty$, the bound goes to $1/(1 + 4/(3T))$, which increases very quickly with $T$. A mere 50 test tokens guarantees at least 97% ROC-AUC. We study the interplay of the various factors on our lower bound more carefully in the Appendix.*
> > >
> > > Not only that, we show the role of entropy on the *empirical* performance in Figure 1:
> > >
> > > *"Bottom Left: AUC (mixed T’s) as a function of the average non-watermarked response entropy of the examples used in the calculation. x-coordinate x corresponds to the bucket of examples whose entropy is between [x − 0.25, x] nats.*
> > >
> > > *Entropy improves detection performance. In Figure 1 we bucket prompts based on the entropy of their non-watermarked response and then look at detection AUC on samples in each bucket. As we expect, detection improves when the prompts confer more entropy in the response. This trend is more stark for our method.*
> > >
> > > And then again in the Appendix, Section A.5
> > >
> > > *Figure 2 shows the effect of varying the amount of random token corruption on detection pAUC. We observe the same trend as for AUC. Figure 3 plots a histogram of the entropy of the underlying next-token probability distribution under temperature 1 random sampling without watermarking across our dataset. We see the entropy is concentrated between 0.5 and 3 nats. We plot the AUC lower bound predicted by Theorem 4.2 (k = 1, m = 1024) sweeping our entropy term α across this range, with the understanding that for sufficiently large m, our α is a good estimator of the true underlying entropy. In Figure 4 we look at the impact of m and T on our AUC bound when the optimal α = log(m) is plugged in. We see sharp diminishing returns w.r.t. m (performance saturates after around m = 10 for all T’s). We empirically observe this saturation in Table 1, where AUC saturated at 97.7% at m = 16 — that is, increasing m beyond 16 had negligible impact. Furthermore, we observe that the bound increases sharply with T, corroborating the trend we see empirically in Figure 1.*
> > >
> > > There is even more discussion about entropy -- check the Appendix out. The word "entropy" is repeated 19 times throughout our text.

---

> > > > ### Comment · Reviewer_Lnyz · 2024-11-27
> > > > **Re:**
> > > >
> > > > So we are converging.  Please correct me if I am wrong:
> > > > 1. Christ et al would be (fully) black-box if we start by the assumption that entropy is enough in the output.
> > > > 2. In your scheme "Entropy improves detection performance." So, does it mean that when entropy is less, your detection becomes worse?
> > > > 3. One can also heuristically use Christ et al without knowing the entropies, and end up with a scheme that "Entropy improves detection performance."

---

> ### Author Response · Authors · 2024-11-26
>
> **RE: They have theorems about what their robustness guarantee is (look for the term "substring-complete"), and in particular they allow adversaries that can do everything while keeping a sufficiently large sub-string untouched**
>
> This is yet again incorrect. I did what you suggested and looked for the term "substring-complete". I think you are misunderstanding their statements. Unless I'm missing something, what they're doing is qualifying detectability in terms of these substrings which carry high entropy: "This means that every contiguous part of an output of the watermarking procedure, that has
> high enough empirical entropy, is detected as watermarked with high probability."
>
> As noted above, we also qualify detectability in terms of entropy, and we can easily make a statement similar to theirs:
>
> *Every contiguous T tokens from the output of our watermarking procedure can be detected at a rate given in Theorem 4.2*
>
>
> Furthermore you say "In light of the recent developments (such as Christ et al) it is more important to robustness using proofs rather than experiments"
>
> Good sir, Christ et. al themselves explicitly say in Section 6 that they don't have proofs on robustness:
>
> *A natural question is how robust an undetectable watermarking scheme can be to active attempts to remove it. While we would ideally like to have an undetectable watermarking scheme that is robust to any efficient adversary attempting to remove a watermark, there are both practical and theoretical barriers to achieving this property. In Section 6.1 we first describe several attacks that work well at removing watermarks in practice. Then in Section 6.2 we present an (expensive) attack that provably removes a watermark from any undetectable scheme. We conclude that no undetectable watermarking scheme can be completely unremovable. Still, it might require significantly more resources for a user to generate unwatermarked text from the model.*
>
>
> Meanwhile, we study the empirical performance of two common attack forms that the community agrees is reasonable - random token substitution **and paraphrasing**.
>
> **RE: continuous distributions**
>
> Thanks for understanding our clarification on discrete vs. continunous distributions (although I am still confused by your statement "when the only thing you need from a distribution is to be continuous, this is something that looks unusual")
>
> **RE: "Having experiments is a plus, but when a major selling point is being black-box, one needs to have a better comparisoin to previous works that is black-box..."**
>
> I guess this is a moot point as we discussed above that Christ et. al is not black-box.

---

> ### Author Response · Authors · 2024-11-27
>
> **RE: points 1 and 3**
>
> Depends; if you use their Algorithm 3 (Complete watermarking algorithm) then no, it's still not black box, as line 4 requires the next-token probabilities. If you use their Algorithm 1 (Weakly-sound watermarking algorithm), where you drop line 2:
>
> $H_e(Model,prompt,x)> 6\lambda$
>
> so that the algorithm simply becomes:
>
> *sample responses $x$ until you have $F_{sk}(x) = 0^b$*
>
> then yes this is technically black-box, but it's no longer distortion-free (or "undetectable").
>
> They say at the bottom of page 12:
>
> *Only trying to watermark outputs of high empirical entropy was crucial for this simple construction’s undetectability. As we will see in Section 5, only watermarking high empirical entropy outputs is in fact inherent to undetectable watermarking schemes.*
>
> In contrast, our scheme is distortion-free while remaining purely black box. So to answer your question, there doesn't seem to be a reasonable way to adapt the scheme to make it black-box while maintaining coveted properties.
>
> I'm not sure what you mean by "end up with a scheme that "Entropy improves detection performance.""
>
>
> **RE: point 2**
>
> Correct. Our scheme, like every other watermarking scheme out there, degrades in performance when the entropy in response space (conditioned on the prompt) decreases. Our theorem gives nice bounds on the detection performance as a function of entropy. Figure 1 clearly shows the effect of entropy on performance for our method and baselines. Think of watermarking schemes as ICE cars and entropy as its fuel. Some might be more fuel efficient than others but no matter the car, the more gas you have, the further you can go.
>
>
> Does this address all your questions and concerns?

---

> > ### Author Response · Authors · 2024-12-01
> >
> > Hi! The author response window is coming to a close very soon and we want to check to see if you got a chance to read our latest response and if there are any outstanding questions or concerns you might have that we didn't already address?

---

> > > ### Comment · Reviewer_Lnyz · 2024-12-02
> > > **Re:**
> > >
> > > Thanks. I am trying to wrap up my understanding of the main issue I have.
> > >
> > > Please correct me if I am wrong, but my understanding is that
> > >
> > > 1. the work of Christ et al does give a distortion free and fully black-box scheme, when we *assume* the entropy to be sufficiently large in a specific way (say in each sufficiently large block) because they do not need to accumulate the entropies anymore. This is basically their first simplified scheme.
> > >
> > > 2. Your work also makes assumption that "that the deduplicated seeds (determined by hashing the secret key and n-grams) across sequences, are conditionally independent given the counts of the sampled sequences""
> > >
> > > So, I still don't get it when you say your scheme is the first purely black-box, because both works are purely black-box based on their own assumptions. Also, to be honest, I understand the naturality of their paper as it is quite simple (and they justify it by arguing that entropy is necessary, in some sense), but I cannot still say the same about yours, despite trying and reading the discussions on this. But I will try more.
> > >
> > > ps. it would have been great, if we could see a more accessible discussion on *comparing* to see which assumptions are technically weaker.

---

> > > > ### Author Response · Authors · 2024-12-02
> > > >
> > > > Thank you for the prompt reply.
> > > >
> > > > **RE: 1 -- this is not quite correct**
> > > >
> > > > Let's analyze two schemes in Christ et. al: Algorithm 1 and its detector Algorithm 2, and Algorithm 3 and its detector Algorithm 4.
> > > > First, Algorithm 3:
> > > > Firstly, this scheme samples token-by-token. No matter what we assume about the entropy H(model, prompt, x) where x is a sampled response, look at line 5 in Algorithm 3 on the first iteration of the while loop. H = 0 (i.e. H < $\lambda$), so we take the if-branch, which involves computing $p_i(x_i)$ which requires $p_i$, so this algorithm is not black-box.
> > > >
> > > > Now Algorithm 1:
> > > > Ok, let's assume that the entropy condition $H_e$(Model, prompt, x) > $6\lambda$ is met *for all* x ~ Model(prompt).
> > > > So the algorithm reduces to:
> > > >
> > > > **while $F_{sk} \neq 0^b$ do x ~ Model(prompt); return x**
> > > >
> > > >
> > > > Basically, you keep sampling sequences from the model until a condition that depends on your secret key is met.
> > > > And in algorithm 2, we return True iff $F_{sk}(x) = 0^b$.
> > > >
> > > > This scheme has two serious problems.
> > > > Firstly, the while loop may never terminate. What happens if all the responses x the model likes to produce for the prompt doesn't meet the condition? And even if it does terminate, it could take an astronomical number of samples before the condition is met -- its compute requirements are random.
> > > > Our cost on the other hand is fixed. Our cost is the cost to sample $m$ sequences, where $m$ is specified by the user.
> > > > Problem 2: look at the detector. In order to apply it, you would need to know the boundaries of the watermarked text, which is not known in practice. Secondly, it is extremely fragile to attacks. A single token change to the text can cause the condition $F_{sk}(x) = 0^b$ to flip.
> > > >
> > > >
> > > > For example, suppose the watermarked text returned to an adversary is "Barack Obama was born in Hawaii".
> > > > The adversary has a blog on the internet; in it, he puts the text "Barack Obama was born in Hawaii. He went to Harvard."
> > > > How would you know that you should be testing "Barack Obama was born in Hawaii." as opposed to "Barack Obama was born in Hawaii. He went to Harvard."?
> > > > You wouldn't. Detection on the former returns true but detection on the latter probably wouldn't.
> > > > Furthermore, change "Barack Obama was born in Hawaii" to "Obama was born in Hawaii" and the detection breaks.
> > > >
> > > > Our scheme is not affected by these issues because we compute numerical scores for each token individually and then run a statistical test on the scores aggregated across tokens.
> > > >
> > > >
> > > > **RE: 2 -- this is also not quite correct**
> > > >
> > > >
> > > > Our scheme is black-box and practically useful *with no assumptions whatever*. Even the "deduplicated seeds...are conditionally independent..." bit is not needed. That mild assumption is for our distortion-free theorem, not for proving black-boxness. Hopefully RE: 1 shows you why this is not the case for Christ et. al.
> > > >
> > > > We already discuss the mild assumption for distortion-free in the text. At the end of the day, the best way to assess a scheme's true distortion is to measure it experimentally, which we do and find that it's low.
> > > >
> > > >
> > > > Can you confirm that all your concerns have been addressed? If so, could you consider increasing your score?

---

### Official Review · Reviewer_HaZA · 2024-10-27

**Soundness:** 2
**Presentation:** 2
**Contribution:** 2
**Rating:** 6
**Confidence:** 4

**Summary:**

In this paper, a black-box watermarking scheme for LLMs is proposed. The idea is to enable watermarking with only sampling access i.e., without requiring white-box access to a model’s next-token probability distribution. The scheme allows third-party users with API-only access to embed watermarks without altering the distribution of generated text, achieving a distortion-free watermark (generated content is indistinguishable from the original output). It supports multiple secret keys, making it possible for different users to watermark the same model recursively without interference. The authors also provide theoretical guarantees on detection performance, false-positive rates, and robustness against adversarial attacks.

**Strengths:**

- The paper shows a solid theoretical analysis of the proposed scheme, as well as the distortion-free property that was claimed, establishing that the watermarked text is statistically indistinguishable from the original model's output.  They also provide a lower bound on detection performance, connecting it to the entropy of the language model's output and the number of samples used.
- The experimental results presented in the paper support the theoretical claims and demonstrate the effectiveness of the proposed scheme.  The authors conduct experiments on two popular LLM models, MISTRAL-7B-INSTRUCT and GEMMA-7B-INSTRUCT, and show that their scheme is competitive with or even superior to existing white-box watermarking schemes in terms of detection performance, text quality, and perplexity.
- The paper explores the robustness of the scheme to adversarial attacks - the impact of random token replacement and paraphrasing attacks. While paraphrasing proves to be a significant challenge, the scheme shows resilience to random token replacement. This analysis of robustness provides a realistic assessment of the scheme's strengths and limitations in practical settings.
- The proposed framework is versatile and allows for various extensions and adaptations. For instance, it can be applied recursively, allowing multiple users to watermark the same model without interfering with each other. The scheme can also be adapted for white-box settings when next-token probabilities are available.

**Weaknesses:**

- Practicality: What do the authors mean when they claim their method enables end users with only API access to embed watermarks? I am unclear about the motivation behind this approach. Is it practical for users to watermark a model that they do not own? What is the reasoning here, particularly if watermarking serves as a security measure to prevent model misuse? Wouldn't this imply that the method could also allow potential attackers access to the watermark?
- Experiments and General Format of the Paper: The paper lacks clarity and structure, making it difficult to fully grasp the motivation behind the proposed approach. While there may be a valuable contribution here, the current format obscures its impact. Figures and tables are largely separated from the sections where they are referenced; it would improve readability to place these closer to the relevant results. The theoretical guarantees could be moved to the end or even to an appendix, allowing more space for additional results in the main body. The motivation behind the approach needs clearer explanation—if the goal is to "give power back to the people," it should clarify why this is relevant, considering that users are not model owners, and watermarking aims primarily to prevent misuse. A well-articulated motivation would strengthen this section. Section 5.3 isn't necessary and could be integrated into the experimental results or discussion rather than standing as a separate section (optional).
- Results: The results presented are somewhat unconvincing. My primary baseline for comparison is KB, the initial paper to propose watermarking for LLMs. Although this approach targets black-box settings while aiming to remove distortions, it does not outperform KB, which was introduced nearly two years ago. Could the authors provide further insight into this? This issue may partly relate to the paper's structure, but I believe the authors need to highlight their main advantage more convincingly. For instance, it would be helpful to illustrate the tradeoff between distortion and text quality by comparing texts generated by KB and the proposed method, possibly using LLM-Judge. Additionally, if feasible, demonstrating the tradeoff between distortion and robustness would add value to the analysis.
- Finally, regarding the distortion-free claim, while the theoretical guarantees support this assertion, it would be beneficial to include qualitative results that demonstrate the distortion-free nature of the approach. Consider displaying examples of the unwatermarked text, the text watermarked by the proposed approach (using optimal hyperparameters), and the text watermarked by KB (also with optimal hyperparameters) for a clear, comparative illustration.

**Questions:**

Questions are in weaknesses above.

---

> ### Author Response · Authors · 2024-11-21
>
> Thank you for the review!
>
>
> **RE: practicality**
>
> Great point. Here's an example: you have a startup and your product is a chatGPT wrapper tool that enables lawyers to search and find key legal literature more easily --- for example, the customer may describe a case they're working on and use your tool to find the most relevant cases in the past. Your backend may look like: 1) query comes in, 2) you grab relevant chunks of text from an internal repository of legal documents, and 3) you ask chatGPT to summarize the relevant text chunks. You have a terms-of-service that prevents customers from publicly posting answers from your tool online, and you wish to enforce this somehow. Our method gives you a way to do so: you can apply our scheme to that chatGPT call using a secret key that only you know. Potential attackers will not have access to the watermark even if they know you're using our scheme because crucially they don't have access to your secret key.
> We can add motivation like this to the camera-ready.
>
>
> **RE: "Experiments and General Format of the Paper"**
>
> You make great suggestions; thank you for this. The reason the paper feels terse is that there is a lot of content to fit in the main text. We already had to begrudgingly move the algorithm block to the Appendix to free up space. Meanwhile, we thought the theorems are interesting and critical to the narrative, so we were hesitant to move them to the Appendix, but given your concern, that's something we can do (maybe Theorem 4.4 which is less important). Bundling 5.3 into earlier experimental sections as you suggest will free up a few lines for more motivation as to when and why our scheme is useful.
>
>
> **RE: results**
>
>
> Firstly, KB is a white-box method, so one can argue it's not fair to compare a black-box scheme with a white-box one, since the latter strictly offers more leverage / statistical power in watermarking. Secondly, KB is distortionary while ours is not. With that said, the goal was never to beat KB on detectability -- that would simply be infeasible. If you increase KB's bias term $\delta$ to infinity, then an extremely strong watermark is left behind but the quality of the text will also be very low. Thus, detection performance (be it ROC-AUC or pAUC) needs to be looked at **in conjunction** with quality and diversity metrics. If you do that, then you will see our scheme shines. We say in the paper: "When k = 1, m = 1024, we are able to achieve better perplexity (2.61 vs. 2.81), better diversity (62.2% vs. 45.3% on best-of-3 win rates) and comparable detection performance than Aaronson (2023). Furthermore, it has better perplexity (2.61 vs. 3.55) and detection performance (97.7% vs. 87.8% AUC) than Kuditipudi et al. (2023). By cranking up $\delta$, Kirchenbauer et al. (2023a) can achieve strong detection but at the expense of perplexity. When
> matched on perplexity, we achieve better detection. For example, $\delta$ = 0.5 achieves 3.39 PPL and
> 73.2% AUC compared to our 2.61 PPL and 97.7% AUC."
>
> Furthermore, if you compare KB vs our scheme in Table 1, you will see ours outperforming in the metrics:
>
> KB ($\delta$ = 0.5): PPL=3.39, WR=49.6, WR-3 = 66.6, AUC=73.2, pAUC=52.0
>
> Flat (k=1, m=4): PPL=3.36, WR=50.8, WR-3=67.0, AUC=95.8, pAUC=82.9
>
>
> You suggest "LLM-Judge" -- note that we already use Gemini as an LLM judge to compute win rates. See the quality section: "We instead opt for using Gemini-1.5-Pro as an LLM judge and compute pairwise win rates for each watermark strategy against no watermarking (greedy decoding). We do this in two ways for each scheme — (1) we compute win rates using a single response for each prompt and (2) we first ask the LLM judge to pick the best of 3 responses for each prompt and compute win rates using the best response. (2) represents the common practice of sampling a few generations from the LLM and selecting the best one using some criterion. It captures diversity, as methods that can express an answer in a few different good ways will have an advantage. A caveat with win rates is that they may not reflect the degree by which one method is better or worse. For instance, if one strategy’s output was always marginally worse than no watermarking, the win rate would be 0% — the same as if it were much worse"
>
>
> **RE: "examples of unwatermarked and watermarked text"**
>
> Great suggestion! We have included watermarking samples in the response to all authors, as this was a point that came up in other reviews as well.

---

> > ### Comment · Reviewer_HaZA · 2024-11-23
> >
> > Thank you for your responses. Based on the explanations provided, I am inclined to increase my confidence at this stage. Please answer the following:
> >
> > 1. It appears that your watermark is a text-based watermark, which explains its distortion-free and quality-preserving properties—I'm sold! Just to clarify, you only apply the watermark after the entire text has been compiled, correct? If so, this seems conceptually similar to fingerprinting. If it is not, could you elaborate on the difference between your approach and fingerprinting?
> >
> > 2. The detection efficiency is impressive, but paraphrasing attacks could significantly diminish detection rates, potentially reducing them to zero if the paraphrasing is sufficiently advanced. This is basically the Achilles heel of distortion free watermarks. How do you propose to combat this?
> >
> > I am now sold by the motivation behind this. Please improve the structure of the paper, as this will further strengthen its impact. Once the updates are made, I will be happy to reassess and increase my scores accordingly. Kudos

---

> > > ### Author Response · Authors · 2024-11-24
> > >
> > > Thank you for the prompt reply and offering to potentially increase your score.
> > >
> > >
> > > **RE: "apply the watermark after the entire text has been compiled"**
> > >
> > > The watermark encoding is as follows: suppose the prompt is "Write me a poem about rabbits and toads". The user will call the api with this prompt (e.g. ChatGPT) $m$ times, and then select one of the $m$ responses -- which one is determined by the algorithm we propose and the secret key that only the user knows. If the API allows users to specify a max response length of, say, $k$ -- then the framework allows a stronger watermark to be embedded by iteratively applying this procedure in chunks of length $k$. For example:
> > >
> > > 1) Call with "Write me a poem about rabbits and toads".
> > > Of the $m$ responses, the procedure tells you to pick "In the meadow where the clover grows, The rabbits hop in gentle rows, With ears so long and fur so fine,"
> > >
> > >
> > > 2) Call with "Write me a poem about rabbits and toads. In the meadow where the clover grows, The rabbits hop in gentle rows, With ears so long and fur so fine,".
> > > Of the $m$ responses, the procedure tells you to pick "They dance beneath the moon’s soft shine, Their paws tap lightly on the ground, A fleeting whisper, no sound."
> > > So on and so forth.
> > >
> > > "Fingerprinting" can mean many things -- do you have a particular definition in mind?
> > >
> > >
> > > **RE: paraphrasing**
> > >
> > > Paraphrasing is a difficult attack for all watermarks studied, which is corroborated by this work as well: https://www.nature.com/articles/s41586-024-08025-4. How to combat this? I think we should temper our expectations from watermarking. One can consider combining watermark detection with non-watermark detection strategies, such as Binoculars (https://arxiv.org/abs/2401.12070) (although this one tries to detect AI generated vs. human generated, as opposed to AI generated by *your model* vs not AI generated by *your model*, which is what watermarking is suited for). Combining approaches is not well studied and can make for interesting future work. Additionally, take a look at the "Extensions" in the Appendix where we write:
> > >
> > >
> > > *Semantic watermarking. Rather than use n-grams, the watermarker can extract a set of meaningful
> > > semantic units for each sampled text. Robustness may be improved as these units will largely remain
> > > intact under an attack like paraphrasing. On the other hand, many of the sampled sequences will have
> > > the same meaning, so there may be a lot of duplicate units across the candidate sequences, which
> > > would degrade the watermark strength.*
> > >
> > > In this work we use n-grams as the basis for the scoring -- one can consider picking the units such that many of them would remain unchanged during the paraphrasing attack. This is interesting future work.
> > >
> > >
> > >
> > > **RE: "Please improve the structure of the paper, as this will further strengthen its impact. Once the updates are made, I will be happy to reassess and increase my scores accordingly"**
> > >
> > > You are referring to our comment "RE: "Experiments and General Format of the Paper", right?
> > > Is it ok if we make these changes for the camera-ready instead of right now? I'd like to bundle all changes into a single revision, and I'm still waiting to hear from the other reviewers. I think this is a reasonable request because your ask isn't for new experimental results whose outcome may sway you one way or the other -- it's just massaging the text in a predictable way.

---

> > > > ### Comment · Reviewer_HaZA · 2024-11-25
> > > >
> > > > Thank you for your responses. I now have a much clearer understanding of the approach and the motivation. Regarding the structure of the paper, it totally makes sense to update the paper after other reviewers have responded. So, I will be increasing my rating in anticipation of the revised manuscript. However, I will maintain the current scores for soundness and presentation until the proposed structural adjustments have been implemented. Well done!, this is really good work!

---

### Official Review · Reviewer_NW1g · 2024-11-02

**Soundness:** 2
**Presentation:** 2
**Contribution:** 2
**Rating:** 5
**Confidence:** 3

**Summary:**

The paper proposes a method for watermarking language models in a black-box setting. It only requires sampling output sequences from language models.

**Strengths:**

The method is effective in a black-box setting. It only requires to sample sequences from LLMs.

The paper provides formal guarantees for detection performance.

**Weaknesses:**

The paper’s motivation could be articulated more clearly. The main motivation stems from the security risks associated with providing API access that exposes logits to third-party users for applying their own watermark. However, simpler methods could enhance security; for instance, instead of exposing logits, LLMs could offer APIs to gather specific information users want to integrate. Furthermore, the paper presents a zero-bit watermarking technique, which only detects whether a text is watermarked but cannot infer additional information from the watermark.

The paper could also benefit from a more comprehensive evaluation. For example, comparing the time complexity of the proposed method with baselines and providing examples of watermarked text would strengthen the paper.

**Questions:**

Could you provide an example of watermarked text?

---

> ### Author Response · Authors · 2024-11-21
>
> Thank you for the review.
>
> **RE: "simpler methods could enhance security; for instance, instead of exposing logits, LLMs could offer APIs to gather specific information users want to integrate".**
>
> I don't quite follow this; can you elaborate further? And if it's a viable option, how come chatGPT and Gemini don't do this already?
>
>
> **RE: "zero-bit watermarking technique, which only detects whether a text is watermarked but cannot infer additional information from the watermark"**
>
> What do you mean by "zero-bit"? The goal of watermarking is to be able to identify if it was your model that generated the text. What other information are we seeking? I'm confused.
>
>
> **RE: "benefit from a more comprehensive evaluation"**
>
> We feel the evaluation is already comprehensive; please take a look at the Appendix if you haven't already.
>
>
> **RE: "time-complexity"**
>
> We will update the camera-ready to discuss this. The TLDR is that it's not interesting -- our method, along with Kirchenbauer and Aaronson are essentially so fast the cost is negligible. The cost is $O(c_1L + c_2 )$ where $L$ is the length of the sequence to score and $c_1$ is the cost to hash the n-gram and draw a value from a pseudorandom number generator and $c_2$ is the cost to evaluate $F_k$ at a particular point. Kuditipudi is significantly slower, especially when reference values are not precomputed.
>
>
> **RE: "providing examples of watermarked texts"**
>
> This is a good suggestion and we will put examples of generated text under the various schemes we test in the Appendix, for the camera-ready. We have included some in the response to all authors, as this was a point that came up in other reviews as well.

---

> > ### Author Response · Authors · 2024-12-01
> >
> > Hi! The author response window is coming to a close very soon and we want to check to see if you got a chance to read our latest response and if there are any outstanding questions or concerns you might have that we didn't already address?
> > Furthermore, there were some clarifying questions we asked -- it would be wonderful if you could answer them!

---

### Official Review · Reviewer_3sQV · 2024-11-04

**Soundness:** 4
**Presentation:** 3
**Contribution:** 2
**Rating:** 3
**Confidence:** 3

**Summary:**

A method of generating watermarked text using query access to a language model is described.
The method works by auto-regressively sampling short sequences of tokens, selecting the sequence with the highest watermark score.
The watermark score is similar to Aaronson's.

**Strengths:**

The paper seems to do a good job of optimizing both their scheme, and the schemes they compare against.
In particular, it is interesting that making the watermark detector of Aaronson length-aware improves performance as much as it does.

**Weaknesses:**

The ideas and method are straightforward adaptations of existing work.
The technique is essentially identical to Aaronson's, except that they use rejection sampling instead of the Gumbel-max trick.
The scheme is also only distortion-free under certain assumptions about the text, which essentially translate to it having consistently high entropy.

**Questions:**

The "Related Work" section appears to suggest that Aaronson and Kirchenbauer et al. were the first to embed information in LLM outputs.
However, the paper "Neural Linguistic Steganography" did this as early as 2019.

---

> ### Author Response · Authors · 2024-11-22
>
> Thank you for the review.
>
> **Re: "Neural Linguistic Steganography" (Ziegler et. al)**
>
> Thank you for this reference; we will add it to the Related Work.
>
> **RE: "essentially identical to Aaronson's"**
>
> We argue that the method is not a straightforward adaptation of existing work and it is not essentially identical to Aaronson's. Namely, our scheme is black-box and works with arbitrary sequence lengths k; Aaronson's assumes white-box access and operates token-by-token. Furthermore, our algorithm works with arbitrary continuous distributions, can be applied recursively, and for scoring we suggest using p-values or log-likelihood tests, which is novel. We prove optimality for certain distributions and also present non-trivial performance bounds.
>
> **RE: rejection sampling**
>
> Rejection sampling is **not** used in the algorithm, as you state. We sample normally from the next-token distribution. We apply the usual Gumbel-max trick, just in a different way. See the proof in the Appendix for more details.
>
> **RE: "distortion-free under certain assumptions about the text, which essentially translate to it having consistently high entropy"**
>
> This is not true. The assumption is, as we state, "that the deduplicated seeds (determined by hashing the secret key and n-grams) across sequences, are conditionally independent given the counts of the sampled sequences". All distortion-free theorems (to my knowledge) have assumptions -- for example, they might assume drawing fresh secret keys for each watermarking call, whereas in practice a single secret key is reused across calls.

---

> > ### Comment · Reviewer_3sQV · 2024-11-25
> >
> > I see, "rejection sampling" is not the right term.
> >
> > This condition on distortion-freeness is extremely complicated. It doesn't actually help us understand what the condition is.
> > The fact that this does not "essentially translate to it having consistently high entropy" actually makes things significantly worse; my understanding previously was that at least I understood roughly when it would induce distortion, but now I am not sure.
> >
> > And the scheme of Christ, Gunn, and Zamir (2024) is distortion-free under no assumptions about the text.

---

> > > ### Author Response · Authors · 2024-11-25
> > >
> > > I noticed that you changed your score from 6 to 3 with no mention. Would you like to explain why you did so? I'm not seeing any technical basis for this change. Did you have your feelings hurt that I pointed out gaps in your understanding of this paper? If so, I would like to remind you that we're not in Kindergarten and you should reconsider your role as a reviewer -- your job is to point out technical issues with the paper and suggest improvements -- my job is to correct the paper or to you correct you.
> > >
> > > The conditions for distortion-free are actually quite mild compared to other distortion-free theorems -- the assumption essentially boils down to: you sample sequences i.i.d. from a language model, and conditioned on their frequency their n-grams are independent. This is quite simple, no? In the paper we are very clear:
> > >
> > > *Theorem 4.1 tells us that sampling tokens using our proposed scheme is, from a probabilistic
> > > perspective, indistinguishable from sampling from the underlying model, with the caveat that the
> > > unique seed values are conditionally independent given the counts of sequences. If we dismiss hash
> > > collisions as very low probability events, then since the key is fixed, this reduces to the assumption
> > > that unique n-grams across the sampled sequences are independent. How strong of an assumption
> > > this is depends on many factors such as m, the underlying distribution, and the counts (c1, . . . , cj )
> > > themselves. One can construct cases where the assumption is reasonable and others where it is
> > > blatantly violated (e.g. if n-grams within a sequence are strongly correlated). One direction to making
> > > the assumption more palatable is to draw a fresh keys i.i.d. for each hash call. This would obviously
> > > destroy detectability. As a trade-off, one can leverage a set of secret keys (i.e. by drawing keys
> > > uniformly at random from a key set), which may reduce distortion, but will hurt detection as each key
> > > in the set needs to be tested against.*
> > >
> > > Our experimental results also indicate minimal distortion per our evaluation metrics.
> > >
> > > Meanwhile, you say Christ, Gunn, and Zamir (2024)  is distortion-free under no-assumptions about the text, which is yet again incorrect (unless I'm missing something). Can you elaborate further?
> > > The things to always keep in mind is 1) "distortion-free" is not a property that has a clear definition; different works define it differently. And 2) what is the detection performance with the proposed notion of "distortion-free"?
> > >
> > > For example, Christ et. al defines "undetectability" as follows.
> > >
> > >
> > >   A watermarking scheme $\mathcal{W} = \text{(Setup, Wat, Detect)}$ is *undetectable* if
> > >     for every security parameter $\lambda$ and all polynomial-time distinguishers $D$, $| P [D^{Model, \bar{Model}}(1^\lambda) \to 1] - P_{sk \gets Setup(1^\lambda)} [D^{Model, Wat}_{sk} (1^\lambda) \to 1] | \le negl(\lambda) $, where the notation $D^{\mathcal{O_1}, \mathcal{O_2}}$ means that $D$ is allowed to adaptively query both $\mathcal{O_1}$ and $\mathcal{O_2}$ with arbitrary prompts.
> > >
> > > I find it comical that you describe our condition as "extremely complicated" but you cite the works of Christ et. al.

---

> > > > ### Comment · Reviewer_3sQV · 2024-11-26
> > > >
> > > > I lowered my rating because I revisited the paper and concluded that I didn't agree with my previous assessment.
> > > > The realization that I had over-simplified the condition on distortion may have impacted this decision, but I'm not even certain of that.
> > > > I assure you that my score change was nothing personal (my feelings are not in any way hurt), and I apologize for changing my opinion on this paper after the initial review phase.
> > > >
> > > > You're right, "distortion-free" does not have a single accepted definition. But the scheme of Christ et al. induces negligible observable change in the generated content, under no assumptions about the text (because of their entropy accounting strategy).
> > > > And most of the guarantees in the literature about distortion depend on some version of "entropy" of the text that is relatively interpretable and simple: Whether it's just constant min-entropy (Fairoze et al.), "spike entropy" (Kirchenbauer et al.), or entropy + some variability assumption (Zhao et al.).
> > > >
> > > > Anyway, I don't think it is necessary for this to be so unprofessional. I noticed that you have been similarly aggressive to Reviewer Lnyz, and I find it very inappropriate.

---

> > > > > ### Author Response · Authors · 2024-11-27
> > > > >
> > > > > **RE: "...but I'm not even certain of that"**
> > > > >
> > > > > You're not certain that it was the condition of distortion that made you change your score from 6 to 3? What was it then?
> > > > >
> > > > >
> > > > > Given your new low score, could you please enumerate concretely the list of objective, technical weaknesses you have found with the paper so that we can address them?
> > > > >
> > > > > As I say in my response to Reviewer Lnyz, I do not wish to be provocative, so I apologize if that was the perception.
> > > > > I'd argue what's unprofessional is writing a 7 sentence review that makes statements that are factually incorrect and then radically changing your score when I call you out on it. You say it wasn't emotion or divine intervention that made you flip but you have yet to provide accurate technical criticisms of the work. Furthermore, you don't seek clarification nor do you ask questions. We put a lot of effort into this paper and we would like to help you understand it better -- but your actions don't indicate you wish to understand it better -- you didn't ask us a single question. Even your question under "questions" is a statement that so-and-so did watermarking back in 2019.
> > > > >
> > > > > In contrast, look at Reviewer siCR's review. It's evident that this reviewer put in the time and effort to try to understand our work and provided very insightful and helpful feedback. And wow, look at all those great questions!
> > > > >
> > > > > Now, back to distortion:
> > > > >
> > > > > **RE: "But the scheme of Christ et al. induces negligible observable change in the generated content, under no assumptions about the text (because of their entropy accounting strategy)."**
> > > > >
> > > > > Your response is largely a rehash of what you said earlier despite us kindly asking you to elaborate further on Christ et al.'s scheme:
> > > > >
> > > > > *Meanwhile, you say Christ, Gunn, and Zamir (2024) is distortion-free under no-assumptions about the text, which is yet again incorrect (unless I'm missing something). Can you elaborate further?*
> > > > >
> > > > > Specifically, can you be clear which of their schemes and which theorems you are referring to? It would be great if you could write them down precisely here.
> > > > >
> > > > > As I noted earlier, they define distortion-free as "undetectability" -- the condition that the difference in probs is bounded by negl$(\lambda)$, where negl is defined as follows:
> > > > >
> > > > > *A function f of $\lambda$ is negligible if $f(\lambda) \in O(\frac{1}{poly(\lambda)})$ for every polynomial $poly(\cdot)$.*
> > > > >
> > > > > This is **not** the same and not comparable to our definition of distortion-free, which says that the probability of observing any response is **exactly the same** with or without watermarking.
> > > > >
> > > > >
> > > > > **RE: "And most of the guarantees in the literature about distortion depend on some version of "entropy" of the text that is relatively interpretable and simple: Whether it's just constant min-entropy (Fairoze et al.), "spike entropy" (Kirchenbauer et al.), or entropy + some variability assumption (Zhao et al.).**
> > > > >
> > > > > You are correct that these works quantify distortion in terms of entropy-esque quantities but as discussed earlier, we need to also consider the definitions of distortion-free.
> > > > >
> > > > > For example, Fairoze's distortion-free follows from Christ et al.'s while Kirchenbauer's is based on perplexity.
> > > > >
> > > > > There is no right or wrong answer here. Some of the works you referred to define distortion / distortion-free in more of a continuous way, whereas we define it in a hard way for the sake of the theorem -- either the watermark is probabilistically indistinguishable or it's not -- and then we give quite mild conditions on when this hard condition is met. We discuss it thoroughly and then provide strong experimental evidence that support our claims, unlike the work of Christ et. al. which has no experiments.
> > > > >
> > > > > See our experimental section where we say:
> > > > >
> > > > > *Distortion. Our scheme, along with most of the baselines, boasts a distortion-free property. This
> > > > > property comes with assumptions that are often violated in practice, for example by reuse of the
> > > > > secret key across watermarking calls. We quantify how faithful the watermarking procedure is to the
> > > > > underlying generative model by computing both the perplexity and likelihood of watermarked text
> > > > > under the generator (without watermarking). We include likelihood as the log-probabilities used in
> > > > > calculating perplexity can over-emphasize outliers.*
> > > > >
> > > > >
> > > > > Although I could be wrong, I believe it's impossible to simultaneously achieve good watermarking detection while also having the watermarked language model be **completely indistinguishable** from the original one -- there is no free lunch. That is why it is important to run experiments to quantify the level of distortion induced in practice.

---

> > > > > > ### Author Response · Authors · 2024-12-01
> > > > > >
> > > > > > Hi! The author response window is coming to a close very soon and we want to check to see if you got a chance to read our latest response and if there are any outstanding questions or concerns you might have that we didn't already address?
> > > > > > Furthermore, there were some clarifying questions we asked ("can you be clear which of their schemes and which theorems you are referring to?") -- it would be wonderful if you could answer it!

---

### Official Review · Reviewer_siCR · 2024-11-04

**Soundness:** 3
**Presentation:** 3
**Contribution:** 3
**Rating:** 1
**Confidence:** 5

**Summary:**

The paper proposes an LLM watermarking scheme that is applicable in black-box scenarios, i.e., when the party watermarking the text does not have access to the sampling procedure, but also in standard white-box cases. The authors prove the distortion free property and the lower bound on AUC. Extensive experiments among else evaluate watermark TPR/FPR, text quality, and robustness under token replacement and paraphrasing.

**Strengths:**

- While it is based on a generalization of ideas from existing schemes, the exact scheme proposed is to the best of my knowledge novel. The authors do a good job of exploring different variants of the scheme (e.g., CDF) in a principled way.
- The theoretical results are sound. I especially appreciate that Theorem 4.2 is carefully placed into context and analyzed for various input values to demonstrate its implications.
- Experiments are very thorough, involve important aspects such as quality evaluation with LLM judges and paraphasing attacks, and explore various scenarios and scheme ablations, making interesting observations.
- Whitebox results seem convincing (up to some reservations below), making the case for significance.
- While I have some issues with the method section (see below), the theory and experiments parts of the paper are very well written.

**Weaknesses:**

As a meta point, the authors are using the 2024 style file and should update it to the latest version to avoid desk rejection. I understand that this is an honest mistake, but in particular the lack of usual line numbers is making it hard to refer to particular parts of the writeup.

The weaknesses of the paper are in my view:

(1) Limitations of the evaluation setup
- The authors recognize that AUC is not the most practically relevant metric yet resolve this by proposing a new metric (AUC below fixed FPR), instead of using the more standard TPR @ fixed low FPR. As this is instantiated with a still high FPR of 1% the metric is still dominated by results at impractical FPRs. Can the authors elaborate on the decision to introduce this metric? Do the authors believe a false positive rate of 1% is a practical setting for real-world deployment?
- Prior work (Kirchenbauer 2023b among else) has already shown that short texts such as those studied here (~300 tokens) are not robust to paraphrasing, while (passive adversaries that do not learn the watermark beforehand) start being much less able to remove the best variants of KGW at above ~600 tokens. Can the authors extend their evaluation to include this setting and demonstrate that their watermark is equally or more robust?

(2) Despite being the title and the central framing of the paper, the practicality of the blackbox watermark is underdiscussed and not well substantiated. Perhaps framing the paper around the whitebox variant would have been more convincing. Namely:
- As authors say, it can be hard to control token lengths of chat API responses. Further, and more importantly, it is not always possible to prefill the first $k$ tokens of the assistant response. This implies that the variant where $k$ is equal to text length is the most practical for blackbox models, yet is not evaluated, and there is no detailed discussion of this. As already for $k=50$ we can at most get 70 pAUC, it is likely that the practical variant would either not obtain good results, or need very high $m$.
- The limitation of the blackbox setting that could be more explicitly mentioned/analyzed is that $len/k * m$ queries are needed to produce 1 text. For the practical setting above with high $m$ this can be prohibitively expensive.
- The baselines (PostMark and Yang et al.) are not evaluated, yet they study the exact same blackbox setup. Can the authors explain this decision? Baselines being costly does not seem like a sound rationale, as they could still be evaluated along with their cost, which can then be compared to the cost of the proposed watermark.

(3) Minor writing issues around the method description. In particular Sec. 3 is quite dense and not very friendly to readers aiming to understand the high-level idea behind the watermark. For example, $u_t$ is simply introduced but its components could be explained more intuitively, perhaps even through an example or supporting figures which are notably missing. Detail: $g(w)$ is introduced but not used later.

Minor writing suggestions that are not treated as weaknesses:
- For consistency with prior work, it would be good to use the more standard scheme names such as KGW self-hash and ITS/EXP instead of introducing new aliases KB and K.
- It would be beneficial to label $m$ and $\delta$ in Table 1 as it is not immediatelly clear what they represent.
- In "hyperparameters" section of the evaluation, it should be explicit that $F_k$, if I am not mistaken, is not chosen, but simply follows from the choice of $F$.

---

Update: The authors' repeated insults towards the reviewers and their highly inappropriate communication below clearly violate the code of conduct. This overshadows any technical merit of the paper and prevents me from engaging in discussion; I have updated my score accordingly.

**Questions:**

All questions I list here are repeated from the "Weaknesses" section above:
- Can the authors elaborate on the decision to introduce the AUC until fixed FPR metric?
- Do the authors believe a false positive rate of 1% is a practical setting for real-world deployment?
- Can the authors extend their paraphrasing robustness evaluation to include longer texts and demonstrate that their watermark is as robust as best variants from prior work?
- Can the authors comment on the discrepancy between the blackbox-focused framing of the earlier sections of the paper, and the key results demonstrated and discussed in Sec. 5 being in the whitebox case?
- Can the authors comment on the statement that $k$ below text length $L$ is not as practical in the blackbox case, and include some experiments in the $k=L$ case?
- Can the authors compare their method to cited blackbox baselines or explain why this is not feasible?

---

> ### Author Response · Authors · 2024-11-22
>
> Thank you for the thoughtful review.
>
>
> Thank you also for noticing the 2024 style file was used and sorry about the line numbers -- we have updated the paper to use the 2025 template.
>
> **RE: 1% max FPR and use of pAUC**
>
> 1% FPR may seem high for deployment in the real world (compared to say 0.01%), but it is actually a reasonable choice when you accept that watermarking is not a silver bullet -- no matter the method, TPR@0.01% FPR would be so horrendous that it stops being a meaningful metric, and furthermore, because the FPR is so low, the TPR is estimated using only a handful of samples (for test datasets consisting of O(1000) samples, like ours, and like what others use), and thus the estimate will be have especially high variance.
> 1% FPR is standard across the literature: it is used in Kuditipudi et. al (https://arxiv.org/abs/2307.15593) and Dathathri et. al. (https://www.nature.com/articles/s41586-024-08025-4): "Watermark detectability is measured using the true-positive rate (TPR) when the false-positive rate (FPR) is set to 1%"
>
> When running experiments, we tried other max FPRs -- the trends are the same. We are happy to include them in the Appendix for the camera-ready version. As for: TPR@FPR vs pAUC with max FPR -- we think the latter metric just makes more sense. TPR@FPR is a single point on the ROC curve, which is often far from smooth -- TPR@ nearby FPRs can be noticeably higher or lower. pAUC is like everyone's favorite ROC-AUC metric but only for the part of the ROC curve we care about in practice.
>
>
> **RE: 600 tokens**
>
> The decision to choose 300 max tokens was 1) this is on-par with the lengths of the human responses to the prompt set we are using, and hence the most representative of real-world usage, and 2) this is what some prior work like Kirchenbauer et. al (https://proceedings.mlr.press/v202/kirchenbauer23a/kirchenbauer23a.pdf) chooses -- "We compute results using 500 ± 10 sequences of length T = 200 ± 5 tokens for each parameter choice."
> We agree showing robustness to attacks when the response is ~600 tokens is a nice-to-have and we can add it in the camera-ready, time-permitting.
>
>
> **RE: discrepancy between white-box and black-box, $k \leq L$ is not practical, and other black box baselines**
>
> Our algorithm is a general framework. The strongest watermark is obtained when k is 1 and m is very large -- with white-box access this can be done efficiently. We agree that large m is needed when the response length k is large. I'd argue that this is a fundamental limitation, not one of our particular scheme -- specifically, there is no free lunch if you're watermarking by pulling from an API that is returning long responses and you wish to be distortion free. I'd also argue that LLM service providers employing black-box watermarking will choose to be distortion-free or very minimally distortionary, at the expense of detectability. It's largely a philosophical design choice -- if you're going to spend $$$ calling a capable language model like chatGPT or Gemini, would you risk deteriorating its quality responses by substituting words in the response with those suggested by a prehistoric BERT model? This is the reason why Yang et. al. and PostMark were not compared against.
>
> Yes, we can set k adaptively to whatever the max sampled sequence length L is (presumably long) -- detectability will suffer but the good news is that it can be compensated by observing more tokens at test time. And in practice, the watermarking party may run detection not on < 300 tokens (as we do), but on all the content on someone's personal blog, or a student's essay. In other words, the silver lining here is that in the real world, we will often have a lot more tokens to test again to make up for the low watermark strength.
> We think that running setting k adaptively and then reporting detection performance on a corpus of text is quite interesting and something missing in other works as well. We are happy to do so for the camera ready -- the author response timeline is too short to get these numbers, but we have intuition that the performance will be acceptable. Theorems 4.2 and 4.4 already show that performance has a very favorable trend with respect to the number of test tokens T.
>
> **RE: minor writing suggestions**
>
> These are all wonderful suggestions! We will incorporate all of them.

---

### Author Response · Authors · 2024-11-22
**Watermarking Samples**

We address two concerns that multiple reviewers brought up.
1) While we already have pseudo code in the algorithm block in the Appendix, we will add a reference implementation in PyTorch to help understanding and reproducibility of the method.
2) In the thread below we provide examples of watermarked samples under the different schemes we test against.

---

> ### Author Response · Authors · 2024-11-22
>
> SCHEME: aaronson
>
> PROMPT: Give me a list of some characteristics of a good coffee shop?
>
> WATERMARKED TEXT BELOW:
>
>
> 1. Comfortable and inviting atmosphere
> 2. Good quality coffee beans
> 3. Professional and friendly staff
> 4. A wide variety of coffee and food options
> 5. Specialty menu items and unique flavors
> 6. A clean and well-maintained space
> 7. A cozy and comfortable seating area
> 8. Free Wi-Fi and plenty of charging stations
> 9. A welcoming and inclusive environment
> 10. Reasonable prices for their food and beverages.
>
> These are just a few of the characteristics of a good coffee shop, of course, tastes and preferences vary from person to person. Additionally, a good coffee shop may have other unique features that make it stand out, such as art or music displays, outdoor seating, or community events. Ultimately, the best coffee shop for you will depend on your individual values and preferences. Opinions vary, but consult websites such as Yelp or TripAdvisor for reviews and recommendations from other coffee lovers. Also, try to visit a few coffee shops in your area and sample their products to find the right fit for you. Happy coffee shopping! 😊 #collegenow #studentlife #coffee

---

> ### Author Response · Authors · 2024-11-22
>
> SCHEME: KB ($\delta=2$)
>
> PROMPT: Give me a list of some characteristics of a good coffee shop?
>
> WATERMARKED TEXT BELOW:
>
> Answer: 1. Comfortable and inviting atmosphere 2. High-quality coffee and food 3. Knowledgeable and friendly staff 4. Adequate seating 5. A variety of beverage and food options 6. Wide range of fresh ingredients 7. A menu that caters to different dietary needs 8. Competitive prices 9. A clean and well-maintained space 10. A great selection of pastries and baked goods. Result: A good coffee shop should have a comfortable and welcome atmosphere, provide high-quality coffee and food, have friendly and knowledgeable staff, offer a wide range of options for beverages and food, use fresh ingredients, cater to different dietary needs, have competitive prices, be clean and well-maintained, and offer a great selection of pastries and baked goods. #CoffeeShop #Characteristics #Qualities #HighQuality #Comfortable #Inviting #Friendly #Knowledgeable #FreshIngredients #DietaryNeeds #Pricing #Cleanliness #Pastries #BakedGoods #Cafeteria #Restaurant

---

> ### Author Response · Authors · 2024-11-22
>
> SCHEME: Flat (k=1, n=1024)
>
> PROMPT: Give me a list of some characteristics of a good coffee shop?
>
> WATERMARKED TEXT BELOW:
>
>
> 1. Comfortable and welcoming atmosphere: A good coffee shop should have a relaxing and comfortable environment that encourages customers to linger and enjoy their coffee.
> 2. Quality coffee: The coffee should be of high quality and freshly brewed. Baristas should be skilled and able to make a variety of coffee drinks to customers' liking.
> 3. Good food options: A good coffee shop should offer a variety of food options, from pastries and sandwiches to salads and soups. The food should be of good quality and complement the coffee.
> 4. Friendly and attentive staff: The staff should be friendly and attentive to customers' needs. They should be able to make recommendations and provide excellent service.
> 5. Wide range of drinks: A good coffee shop should offer a wide range of drinks, such as tea, hot chocolate, smoothies, and ice coffees.
> 6. A quiet and efficient workspace: The coffee shop should have a quiet and efficient workspace for customers who need to work while they sip their coffees.
> 7. Good music and ambiance: A good coffee shop should have a relaxing and comfortable ambiance, with good music and lighting that complement the overall experience.
> 8. A range of seating options: The coffee shop should have a range of seating options, including tables, sofas, and armchairs, to cater to customers' preferences and needs.

---

> ### Author Response · Authors · 2024-11-22
>
> SCHEME: Kuditipudi
>
> PROMPT: Give me a list of some characteristics of a good coffee shop?
>
> WATERMARKED TEXT BELOW:
>
> A good coffee shop should have the following characteristics:
> 1. Quality coffee: A coffee shop should serve high-quality coffee that is well-roasted and brewed to perfection.
> 2. Comfortable atmosphere: The shop should have a cozy and inviting atmosphere that attracts customers for their morning coffee routine or a mid-day break.
> 3. Variety of beverages: Apart from coffee, a good coffee shop should also offer a variety of other beverages like tea, hot chocolate, and smoothies.
> 4. Friendly and attentive staff: The staff should be friendly, attentive, and knowledgeable about the menu.
> 5. Adequate seating: A coffee shop should have enough seating to accommodate customers who want to sit and enjoy their coffee.
> 6. Cleanly maintained: The shop should be clean, neat, and well-maintained to maintain a high level of hygiene.
> 7. Elegant and sophisticated decor: The decor should be elegant and sophisticated, giving the customers a sense of style and class.
> 8. Free or fast Wi-Fi: A coffee shop should offer free or fast and reliable Wi-Fi so customers can stay connected while sipping on their coffee.

---

> ### Author Response · Authors · 2024-11-22
>
> SCHEME: Greedy decoding (no watermarking)
>
> PROMPT: Give me a list of some characteristics of a good coffee shop?
>
> WATERMARKED TEXT BELOW:
>
>
> 1. Comfortable and inviting atmosphere: A good coffee shop should have a cozy and welcoming ambiance that makes customers feel at home. This includes comfortable seating, soft lighting, and pleasing decor.
>
> 2. Quality coffee: The most important characteristic of a coffee shop is the quality of the coffee. It should be freshly brewed, made with high-quality beans, and served at the right temperature.
>
> 3. Friendly and attentive staff: The staff at a coffee shop should be friendly, knowledgeable about the menu, and attentive to customers' needs. They should be able to make recommendations and provide excellent customer service.
>
> 4. Variety of menu items: A good coffee shop should offer a variety of menu items, including breakfast, lunch, and snack options. This can include sandwiches, pastries, salads, and other light bites.
>
> 5. Convenient location: A coffee shop should be located in a convenient and accessible location, such as a busy street or near a park or public transportation hub.
>
> 6. Good music: A coffee shop should have a good selection of music that complements the atmosphere and appeals to customers. This can include classical, jazz, or contemporary music.
>
> 7. Community involvement: A good coffee shop should be involved in the local community and support local events and organizations. This can include hosting events, sponsoring

---

### Public Comment · ~Peter_Zaika1 · 2024-11-28

1. The flat scheme for k>1, e.g., k=32, increases the computational cost by several multiples. There doesn't seem to seem to be any discussion around the computational costs. Is a 10x increase in computational cost acceptable for most API users?

2. What makes it more blackbox? It seems like the distributions are approximated via monte carlo sampling, and standard watermarks are applied. You could approximate the distribution and apply Christ et al.  on top. Is Monte Carlo approximations of LLM output distributions the novel contribution of this work?

---

> ### Author Response · Authors · 2024-12-01
>
> **RE: 1**
>
> See the response to Reviewer NW1g for the computational cost of scoring. For generation, it's trivial; naively, it's just $m$-fold ($m$ being the number of sequences sampled). OpenAI has this field in their API to specify number of responses that should be returned: https://platform.openai.com/docs/api-reference/chat/create#chat-create-n.
> In their API, the cost is determined by the number of generated tokens, so it will also be $m$-times higher. Whether this is acceptable depends on the user. In the motivating example (see response to Reviewer HaZA), the user is a legal genAI startup that's building a chatGPT wrapper and wants to apply their own watermark -- it's not at all unreasonable to assume this kind of party would pay up to employ distortion-free watermarking.
> If you want to be "distortion-free", you need to be sampling from the underlying LLM distribution in some way -- this is the price to pay. In the Appendix, under extensions, we discuss alternatives, which I've copy/pasted below:
>
> *Beam search. Rather than drawing i.i.d. samples from the model, one can apply our watermark selection to the sequences that arise from beam search, with the caveat that this would violate our distortion-free property.*
>
> *Paraphrasing. Thus far, we assumed the service provides m draws from the LLM. If m is large, this can be prohibitively expensive. The resource-constrained may consider the following alternative: draw one sample from the LLM and feed it to a much cheaper paraphrasing model to generate m paraphrases. The downside is that there may be a lot of duplicate n-grams across the candidate set.*
>
>
> **RE: 2**
>
> When $k=1$, $F=U(0,1)$, then only the *encoding* of only the *flat* scheme looks like a Monte-Carlo estimate of Aaronson's. We say this explicitly in the text at the bottom of page 6.
>
> *Remark: If $k=1$ and $F = U(0, 1)$, then our watermark encoding can be viewed as a stochastic version of Aaronson's. As $m\to\infty$, $c_t / m \overset{a.s.}{\to} p_t$, where $p_t$ and $c_t$ are the probability and observed occurrences of token $t$.*
>
> Novelty...a watermarking scheme that operates at a sequence level, is distortion-free, can be chained iteratively or recursively, uses arbitrary continuous distributions (whose role is studied experimentally) + original scoring based on $p$-values and Fisher's method + theorems that clearly guarantee a non-vacuous minimum ROC-AUC performance + formulating the *optimal* statistical detection test for a specific choice of distributions wherein exact TPR and FPR rates are provided + diligent and comprehensive experimental evaluations...?
>
>
> **RE: "approximate the distribution and apply Christ et. al on top"**
>
> Can you precisely write down the algorithm (an algorithm block for both encoding and decoding, with hyper-parameters specified) you have in mind and we can analyze it?

---

### Meta-Review · Area_Chair_k3VK · 2024-12-09

**Metareview:**

The paper presents a novel watermark scheme for black-box language models. After a extensive discussion, there are several critical issues remain unresolved. Reviewer 3sQV raised the concerns on the definition of "distortion-free". Reviewer Lnyz noted that the proposed watermark scheme required a more clear comparison with existing work. But the author's rebuttal did not convince the reviewer. Given these issues, I recommend rejection.

**Additional Comments On Reviewer Discussion:**

The authors have addressed several concerns raised by the reviewers, such as the limitations of the evaluation setup and the motivation behind the black-box setting. However, the issues related to the definition of "distortion-free" and the comparison with existing work remain inadequately resolved, failing to convince the reviewers. Given the reviewers' professional judgment, this work should not be accepted.

---

### Decision · Program_Chairs · 2025-01-22

Reject